# CIC: Contrastive Intrinsic Control for Unsupervised Skill Discovery

## Abstract

We introduce Contrastive Intrinsic Control (CIC) - an algorithm for unsupervised skill discovery that maximizes the mutual information between skills and state transitions. In contrast to most prior approaches, CIC uses a decomposition of the mutual information that explicitly incentivizes diverse behaviors by maximizing state entropy. We derive a novel lower bound estimate for the mutual information which combines a particle estimator for state entropy to generate diverse behaviors and contrastive learning to distill these behaviors into distinct skills. We evaluate our algorithm on the Unsupervised Reinforcement Learning Benchmark, which consists of a long reward-free pre-training phase followed by a short adaptation phase to downstream tasks with extrinsic rewards. We find that CIC improves on prior unsupervised skill discovery methods by $91\%$ and the next-leading overall exploration algorithm by $26\%$ in terms of downstream task performance.

## 1 Introduction

Deep Reinforcement Learning (RL) is a powerful approach toward solving complex control tasks in the presence of extrinsic rewards. Successful applications include playing video games from pixels (Mnih et al., 2015), mastering the game of Go (Silver et al., 2017; 2018), robotic locomotion (Schulman et al., 2016; 2017; Peng et al., 2018) and dexterous manipulation (Rajeswaran et al., 2018; OpenAI, 2018; 2019) policies. While effective, the above advances produced agents that are unable to generalize to new downstream tasks beyond the one they were trained to solve. Humans and animals on the other hand are able to acquire skills without supervision and apply them efficiently to a variety of downstream tasks. In this work, we seek to train agents that acquire skills without supervision with generalization capabilities by efficiently adapting these skills to downstream tasks.

Over the last few years, unsupervised RL has emerged as a promising framework for developing RL agents that can generalize to new tasks. In the unsupervised RL setting, agents are first pre-trained with self-supervised intrinsic

$$I(\tau; z) = \mathcal{H}(\tau) - \mathcal{H}(\tau|z)$$

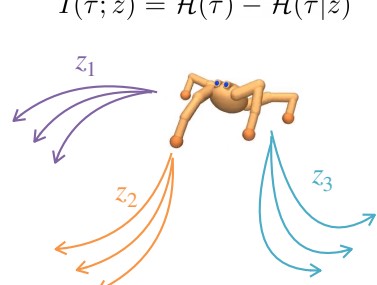

Figure 1: This work deals with unsupervised skill discovery through mutual information maximization. We introduce Contrastive Intrinsic Control (CIC) – a new unsupervised RL algorithm that explores and adapts more efficiently than prior methods.

rewards and then finetuned to downstream tasks with extrinsic rewards. Unsupervised RL algorithms broadly fall into three categories - knowledge-based, data-based, and competence-based methods[1]. Knowledge-based methods maximize the error or uncertainty of a predictive model (Pathak et al., 2017; 2019; Burda et al., 2019b). Data-based methods maximize the entropy of the agent's visitation (Liu & Abbeel, 2021a; Yarats et al., 2021b). Competence-based methods learn skills that generate diverse behaviors (Eysenbach et al., 2019; Gregor et al., 2017). This work falls into the latter category of competence-based methods.

---

[1]These categories for exploration algorithms were introduced by Srinivas & Abbeel (2021) and inspired by Oudeyer et al. (2007).

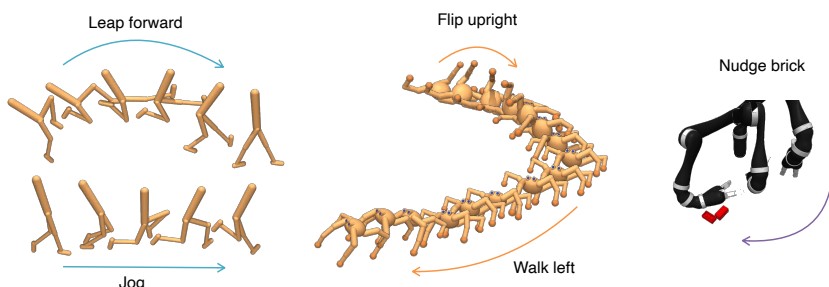

Figure 2: Qualitative visualizations of unsupervised skills discovered in Walker, Quadruped, and Jaco arm environments. The Walker learns to balance and move, the Quadruped learns to flip upright and walk, and the 6 DOF robotic arm learns how to move without locking. Unlike prior competence-based methods for continuous control which evaluate on OpenAI Gym (e.g. Eysenbach et al. (2019)), which reset the environment when the agent loses balance, CIC is able to learn skills in fixed episode length environments which are much harder to explore (see Appendix K).

Unlike knowledge-based and data-based algorithms, competence-based algorithms simultaneously address both the exploration challenge as well as distilling the generated experience in the form of reusable skills. This makes them particularly appealing, since the resulting skill-based policies (or skills themselves) can be finetuned to efficiently solve downstream tasks. While there are many self-supervised objectives that can be utilized, our work falls into a family of methods that learns skills by maximizing the mutual information between visited states and latent skill vectors. Many earlier works have investigated optimizing such objectives (Eysenbach et al., 2019; Gregor et al., 2017; Kwon, 2021; Sharma et al., 2020). However, competence-based methods have been empirically challenging to train and have under-performed when compared to knowledge and data-based methods (Laskin et al., 2021).

In this work, we take a closer look at the challenges of pre-training agents with competence-based algorithms. We introduce Contrastive Intrinsic Control (CIC) – an exploration algorithm that uses a new estimator for the mutual information objective. CIC combines particle estimation for state entropy (Singh et al., 2003; Liu & Abbeel, 2021a) and noise contrastive estimation (Gutmann & Hyvärinen, 2010) for the conditional entropy which enables it to both generate diverse behaviors *(exploration)* and discriminate high-dimensional continuous skills *(exploitation)*. To the best of our knowledge, CIC is the first exploration algorithm to utilize noise contrastive estimation to discriminate between latent skill vectors. Empirically, we show that CIC adapts to downstream tasks more efficiently than prior exploration approaches on the Unsupervised Reinforcement Learning Benchmark (URLB). CIC achieves 91% higher returns on downstream tasks than prior competence-based algorithms and 26% higher returns than the next-best exploration algorithm overall.

## 2 BACKGROUND AND NOTATION

**Markov Decision Process:** We operate under the assumption that our system is described by a Markov Decision Process (MDP) (Sutton & Barto, 2018). An MDP consiss of the tuple $(\mathcal{S}, \mathcal{A}, \mathcal{P}, r, \gamma)$ which has states $s \in \mathcal{S}$, actions $a \in \mathcal{A}$, transition dynamics $p(s'|s,a) \sim \mathcal{P}$, a reward function $r$, and a discount factor $\gamma$. In an MDP, at each timestep $t$, an agent observes the current state $s$, selects an action from a policy $a \sim \pi(\cdot|s)$, and then observes the reward and next state once it acts in the environment: $r, s' \sim \text{env.step}(a)$. Note that usually $r$ refers to an extrinsic reward. However, in this work we will first be pre-training an agent with intrinsic rewards $r^{\text{int}}$ and finetuning on extrinsic rewards $r^{\text{ext}}$.

For convenience we also introduce the variable $\tau(s)$ which refers to any function of the states $s$. For instance $\tau$ can be a single state, a pair of states, or a sequence depending on the algorithm. Our method uses $\tau = (s, s')$ to encourage diverse state transitions while other methods have different specifications for $\tau$. Importantly, $\tau$ does not denote a state-action trajectory, but is rather shorthand for any function of the states encountered by the agent. In addition to the standard MDP notation, we will also be learning skills $z \in \mathcal{Z}$ and our policy will be skill-conditioned $a \sim \pi(\cdot|s,z)$.

**Unsupervised Skill Discovery through Mutual Information Maximization:** Most competence-based approaches to exploration maximize the mutual information between states and skills. Our

work and a large body of prior research (Eysenbach et al., 2019; Sharma et al., 2020; Gregor et al., 2017; Achiam et al., 2018; Lee et al., 2019; Liu & Abbeel, 2021b) aims to maximize a mutual information objective with the following general form:

$$I(\tau; z) = \mathcal{H}(z) - \mathcal{H}(z|\tau) = \mathcal{H}(\tau) - \mathcal{H}(\tau|z) \tag{1}$$

Competence-based algorithms use different choices for $\tau$ and can condition on additional information such as actions or starting states. For a full summary of competence-based algorithms and their objectives see Table 1 in Appendix D.

**Lower Bound Estimates of Mutual Information:** The mutual information $I(s; z)$ is intractable to compute directly. Since we wish to maximize $I(s; z)$, we can approximate this objective by instead maximizing a lower bound estimate. Most known mutual information maximization algorithms use the variational lower bound introduced in Barber & Agakov (2003):

$$I(\tau; z) = \mathcal{H}(z) - \mathcal{H}(z|\tau) \geq \mathcal{H}(z) + \mathbb{E}[\log q(z|\tau)] \tag{2}$$

Note that the variational lower bound can be applied to both decompositions of the mutual information. The design decisions of a competence-based algorithm therefore come down to (i) which decomposition of $I(\tau; z)$ to use, (ii) whether to use discrete or continuous skills, (iii) how to estimate $H(z)$ or $H(\tau)$, and finally (iv) how to estimate $H(z|\tau)$ or $H(\tau|z)$.

## 3 MOTIVATION

The results from the recent Unsupervised Reinforcement Learning Benchmark (URLB) introduced in Laskin et al. (2021), suggest that pre-training with competence-based approaches underperforms relative to knowledge-based and data-based baselines on DeepMind Control (DMC). We argue that the underlying issue with current competence-based algorithms when deployed on harder exploration environments like DMC has to do with the currently used estimators for $I(\tau; z)$ rather than the objective itself. To produce structured skills that lead to diverse behaviors, $I(\tau; z)$ estimators must (i) explicitly encourage diverse behaviors and (ii) have the capacity to discriminate between high-dimensional continuous skills. Current approaches do not satisfy both criteria.

*Competence-base algorithms do not ensure diverse behaviors:* Most of the best known competence-based approaches (Eysenbach et al., 2019; Gregor et al., 2017; Achiam et al., 2018; Lee et al., 2019), optimize the first decomposition of the mutual information $\mathcal{H}(z) - \mathcal{H}(z|\tau)$. The issue with this decomposition is that while it ensures diversity of skill vectors it does not ensure diverse behavior from the policy, meaning $\max \mathcal{H}(z)$ does not imply $\max \mathcal{H}(\tau)$. Of course, if $H(z) - \mathcal{H}(z|\tau)$ is maximized and the skill dimension is sufficiently large, then $\mathcal{H}(\tau)$ will also be maximized implicitly. Yet in practice, to learn an accurate discriminator $q(z|\tau)$, the above methods assume skill spaces that are much smaller than the state space (see Table 1), and thus behavioral diversity may not be guaranteed. In contrast, the decomposition $I(\tau; z) = \mathcal{H}(\tau) - \mathcal{H}(\tau|z)$ ensures diverse behaviors through the entropy term $\mathcal{H}(\tau)$. Methods that utilize this decomposition include Liu & Abbeel (2021b); Sharma et al. (2020).

*Why it is important to utilize high-dimensional skills:* Once a policy is capable of generating diverse behaviors, it is important that the discriminator can distill these behaviors into distinct skills. If the set of behaviors outnumbers the set of skills, this will result in degenerate skills – when one skill maps to multiple different behaviors. It is therefore important that the discriminator can accommodate continuous skills of sufficiently high dimension. Empirically, the discriminators used in prior work utilize only low-dimensional continuous skill vectors. DIAYN (Eysenbach et al., 2019) utilized 16 dimensional skills, DADS (Sharma et al., 2020) utilizes continuous skills of dimension $2 - 5$, while APS (Liu & Abbeel, 2021b), an algorithm that utilizes successor features (Barreto et al., 2016; Hansen et al., 2020) for the discriminator, is only capable of learning continuous skills with dimension 10. We show how small skill spaces can lead to ineffective exploration in a simple gridworld setting in Appendix I and evidence that skill dimension affects performance in Fig. 6.

*On the importance of benchmarks for evaluation:* While prior competence-based approaches such as DIAYN (Eysenbach et al., 2019) were evaluated on OpenAI Gym (Brockman et al., 2016), Gym environment episodes terminate when the agent loses balance thereby leaking some aspects of extrinsic signal to the exploration agent. On the other hand, DMC episodes have fixed length. We show

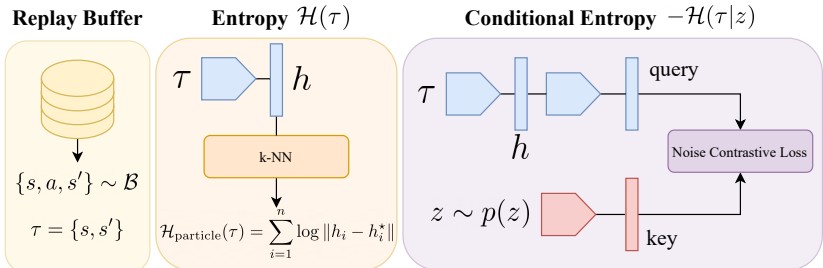

Figure 3: Architecture illustrating the practical implementation of CIC . During a gradient update step, random $\tau = (s, s')$ tuples are sampled from the replay buffer, then a particle estimator is used to compute the entropy and a noise contrastive loss to compute the conditional entropy. The contrastive loss is backpropagated through the entire architecture. The entropy and contrastive terms are then scaled and added to form the intrinsic reward. The RL agent is optimized with a DDPG Lillicrap et al. (2016).

in Appendix K that this small difference in environments results in large performance differences. In Fig. 11 we show that DIAYN is able to learn diverse skills in Gym but not in DMC, which is consistent with both observations from DIAYN and URLB papers. Due to fixed episode lengths, DMC tasks are harder for reward-free exploration since agents must learn to balance without supervision.

## 4 METHOD

### 4.1 THE CIC ESTIMATOR

From Section 3 we are motivated to find an estimator for $I(\tau; z)$ that explicitly maximizes the entropy $\mathcal{H}(s)$ through the second decomposition $I(\tau; z) = \mathcal{H}(\tau) - \mathcal{H}(\tau|z)$. We also desire that our method's discriminator is capable of supporting high-dimensional continuous skills to ensure maximal behavioral diversity.[2] Note that $\tau$ is not a trajectory but some function of states.

In this work, we propose a new estimator for $I(\tau; z)$ which combines the use of a particle estimator for the entropy (Liu & Abbeel, 2021a) and noise contrastive estimation (Gutmann & Hyvärinen, 2010) for the conditional entropy. Our proposed sample-based estimator is:

$$F_{\text{CIC}}(\tau_i, z_i) := \mathcal{H}_{\text{particle}}(\tau_i) + \mathbb{E}\left[f(\tau_i, z_i) - \log \frac{1}{N} \sum_{j=1}^{N} \exp(f(\tau_j, z_i))\right] \quad (3)$$

where $N$ is the number of samples, $\tau = (s, s')$, and $\mathcal{H}_{\text{particle}}(\tau)$ is a particle estimator (Singh et al., 2003; Beirlant, 1997; Liu & Abbeel, 2021a) which estimates entropy by computing the distance between each particle $h_i$ and its $k$-th nearest neighbor $h_i^\star$ such that $\mathcal{H}_{\text{particle}}(\tau) \propto \sum_{i=1}^{n} \log \|h_i - h_i^\star\|$. The CIC estimator should achieve the best of both worlds – encouraging exploration through $\max H(\tau)$ and distilling behaviors into skills through contrastive representation learning. We first show that Eq. 3 is a valid lower bound for $I(\tau; z)$.

**Theorem 1.** *Let $F_{CIC}(\tau, z)$ be defined as in Eq. 3, we have that $F_{CIC}(\tau, z)$ is a lower bound of the mutual information: $I(\tau, z) \geq F_{CIC}(\tau, z)$, where $f(\tau, z)$ is any real function of $\tau$ and $z$.*

*Proof.* First we find a variational lower bound for $I(\tau; z)$ where the inequality is due to Barber & Agakov (2003).

$$I(\tau; z) = \mathcal{H}(\tau) - \mathcal{H}(\tau|z) \geq \mathcal{H}(\tau) + \mathbb{E}[\log q(\tau|z)], \quad (4)$$

From Contrastive Predictive Coding (CPC) (Oord et al., 2018) we can also have a sample based lower bound for $I(\tau; z)$.

---

[2]In high-dimensional state-action spaces the number of distinct behaviors can be quite large.

$$I(\tau; z) \geq F_{\text{CPC}}(\tau_i, z_i) := \mathbb{E}\left[ f(\tau_i, z_i) - \log \frac{1}{N} \sum_{j=1}^{N} \exp(f(\tau_j, z_i)) \right]. \tag{5}$$

As shown in Oord et al. (2018), this bound is upper bound by $\log N$ which means the bound will be loose when $I(\tau; z) \geq \log N$. To overcome this limitation, we note that we can also parameterize the variational density in Eq. 4 with a noise contrastive estimator $q(\tau_i|z_i) = \exp\left(f(\tau_i, z_i)\right) / \left(\frac{1}{N} \sum_{j=1}^{N} \exp(f(\tau_j, z_i))\right)$. We therefore have $I(\tau_i; z_i) \geq \mathcal{H}(\tau_i) + F_{\text{CPC}}(\tau_i, z_i)$ which completes the proof. $\square$

A favorable property of the CIC estimator is that it provides a tighter lower bound than CPC for mutual information $I(\tau; z) \geq F_{\text{CIC}}(\tau, z) \geq F_{\text{CPC}}(\tau, z)$. In contrast to the CPC estimator, CIC is more suitable for exploration due to the explicit presence of $\mathcal{H}(\tau)$, which helps learning meaningful representations and behaviors as evident in recent work (Campos et al., 2020; Mutti et al., 2021; Liu & Abbeel, 2021a; Campos et al., 2021a; Yarats et al., 2021b) whereas Eq. 5 does not explicitly encourage exploration. On the other hand, contrastive learning has been demonstrated as a powerful approach for representation learning in vision and reinforcement learning (Chen et al., 2020; Oord et al., 2018; Laskin et al., 2020b). It is therefore interesting to combine these two objectives into a single intrinsic reward.

## 4.2 INTRINSIC REWARD AND INTERPRETATION OF CIC ESTIMATION

*Intrinsic Reward:* We parameterize $f(\tau, z) = g_{\psi_1}(\tau)^\top g_{\psi_2}(z)$ where $\tau = (s, s')$ is a transition tuple and $g_{\psi_k}$ are neural encoders. This inner product is similar to the one used in the SimCLR (Chen et al., 2020) representation learning loss. We then use a particle estimator (Singh et al., 2003; Beirlant, 1997) as in Liu & Abbeel (2021a) for the entropy term. Similar to Liu & Abbeel (2021a); Yarats et al. (2021b) rather than using the exact form of the particle estimator we estimate the entropy up to entropy up to a proportionality constant, and therefore introduce a hyperparameter $\alpha$ to weigh the entropy and CPC terms. With this parametrization the intrinsic reward for the unsupervised RL agent takes on the following form:

$$r^{\text{int}}(\tau_i, z_i) := \alpha \log \left[ \frac{1}{N_k} \sum_{h_i^\star \in N_k} \|h_i - h_i^\star\| \right] + (1 - \alpha) \left[ f(\tau_i, z_i) - \log \frac{1}{N} \sum_{j=1}^{N} \exp(f(\tau_j, z_i))) \right] \tag{6}$$

where $h_i$ is an embedding of $\tau_i$ shown in Fig. 3, $h_i^*$ is a kNN embedding, $N_k$ is the number of kNNs, and $N - 1$ is the number of negatives. The total number of elements in the summation is $N$ because it includes one positive.

*Explore and Exploit:* We can interpret the two terms of Eq. 6 as contributing two different behaviors to the exploration algorithm. The entropy term $\mathcal{H}(\tau)$ encourages exploration by maximizing state diversity. The variational density $q(\tau|z)$ encourages exploitation by ensuring that skills $z$ lead to predictable states $\tau$. Together the two terms form an intrinsic reward that incentivizes diverse yet predictable behavior from the RL agent.

*Asymptotic Behavior of the Intrinsic Reward:* When maximum entropy is reached the $\mathcal{H}(\tau)$ term in Eq. 6 will vanish since there are no new states to discover. Therefore asymptotically exploration will stop. However, the variational density $q(\tau|z)$ parameterized by CPC will continue distilling all states in the environment into skills $z$ until they are maximally distinct such that $\mathcal{H}(\tau|z) = \varepsilon \ll 1$.

## 5 PRACTICAL IMPLEMENTATION

Our practical implementation consists of two main components: the RL optimization algorithm and the architecture for specifying the intrinsic reward. For fairness and clarity of comparison, we use the same RL optimization algorithm for our method and all baselines in this work. Since the baselines

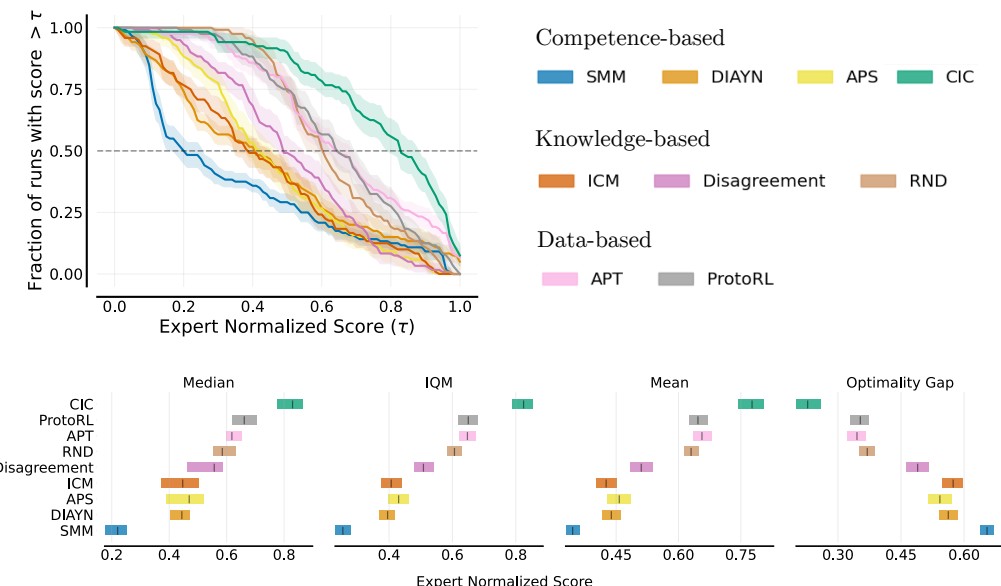

Figure 4: We report the aggregate statistics using stratified bootstrap intervals (Agarwal et al., 2021) for 12 downstream tasks on URLB with 10 seeds, so each statistic for each algorithm has 120 seeds in total. We find that overall, CIC achieves leading performance on URLB in terms of the IQM, mean, and OG statistics. As recommended by Agarwal et al. (2021), we use the IQM as our primary performance measure. In terms of IQM, CIC improves upon the next best skill discovery algorithm (APS) by $91\%$ and the next best algorithm overall (ProtoRL) by $26\%$.

implemented in URLB (Laskin et al., 2021) use a DDPG[3] (Lillicrap et al., 2016) as their backbone, we opt for the same DDPG architecture to optimize our method as well (see Appendix B). For the full algorithm

*Architecture for intrinsic rewards:* We use a particle estimator as in Liu & Abbeel (2021a) to estimate $\mathcal{H}(s)$. To compute the variational density $q(\tau|z)$, we first sample skills from uniform noise $z \sim p(z)$ where $p(z)$ is the uniform distribution over the $[0, 1]$ interval. We then use two MLP encoders to embed $g_{\psi_1}(\tau)$ and $g_{\psi_2}(z)$, and optimize the parameters $\psi_1, \psi_2$ with the CPC loss similar to SimCLR (Chen et al., 2020) since $f(\tau, z) = g_{\psi_1}(\tau)^T g_{\psi_2}(z)$. We fix the hyperparameters across all domains and downstream tasks. We refer the reader to the Appendices E and F for the full algorithm and a full list of hyperparameters.

*Adapting to downstream tasks:* To adapt to downstream tasks we follow the same procedure for competence-based method adaptation as in URLB (Laskin et al., 2021). During the first 4k environment interactions we populate the DDPG replay buffer with samples and use the extrinsic rewards collected during this period to finetune the skill vector $z$. While it's common to finetune skills with Cross Entropy Adaptation (CMA), given our limited budget of 4k samples (only 4 episodes) we find that a simple grid sweep of skills over the interval $[0, 1]$ produces the best results (see Fig. 6). After this, we fix the skill $z$ and finetune the DDPG actor-critic parameters against the extrinsic reward for the remaining 96k steps. Note that competence-based methods in URLB also finetune their skills during the first 4k finetuning steps ensuring a fair comparison between the methods. The full adaptation procedure is detailed in Appendix E.

## 6 EXPERIMENTAL SETUP

**Environments** We evaluate our approach on tasks from URLB, which consists of twelve downstream tasks across three challenging continuous control domains for exploration algorithms – walker, quadruped, and Jaco arm. Walker requires a biped constrained to a 2D vertical plane to per-

---

[3]It was recently was shown that a DDPG achieves state-of-the-art performance (Yarats et al., 2021a) on DeepMind Control (Tassa et al., 2018) and is more stable than SAC (Haarnoja et al., 2018) on this benchmark.

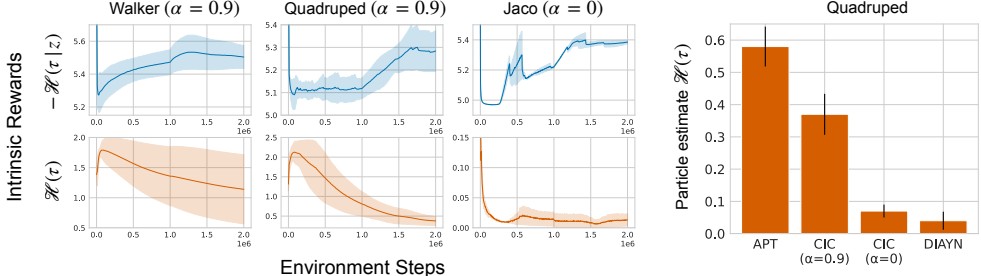

Figure 5: We visualize the contributions to the CIC intrinsic reward from the entropy and disciminator terms across the three URLB domains. Since $\mathcal{H}(\tau)$ and $\mathcal{H}(\tau|z)$ terms are on different scales we set the hyperparameter $\alpha = 0.9$ to weight the two terms equally for the Walker and Quadruped tasks. For Jaco, we find that discriminator-only CIC ($\alpha = 0.0$) is sufficient because random exploration results in meaningful behaviors since the arm is fixed to the table and therefore can't fall. While the CPC intrinsic reward increases throughout training, the entropy reward decreases and settles at a non-zero value. Without explicit entropy maximization, the entropy of CIC approaches zero. Compared to APT, CIC achieves smaller entropy as expected since the discriminator counteracts the entropy term. Compared to DIAYN, CIC achieves substantially higher entropy.

form locomotion tasks while balancing. Quadruped is more challenging due to a higher-dimensional state-action space and requires a quadruped to in a 3D environment to learn locomotion skills. Jaco arm is a 6-DOF robotic arm with a three-finger gripper to move and manipulate objects without locking. All three environments are challenging in the absence of an extrinsic reward.

**Baselines:** We compare CIC to baselines across all three exploration categories. Knowledge-based basedlines include ICM (Pathak et al., 2017), Disagreement (Pathak et al., 2019), and RND (Burda et al., 2019b). Data-based baselines incude APT (Liu & Abbeel, 2021a) and ProtoRL (Yarats et al., 2021b). Competence-based baselines include DIAYN Eysenbach et al. (2019), SMM Lee et al. (2019), and APS (Liu & Abbeel, 2021b). The closest baselines to CIC are APT, which is similar to CIC with $\alpha = 1.0$ (no discriminator), and APS which uses the same decomposition of mutual information as CIC and also uses a particle entropy estimate for $\mathcal{H}(\tau)$. The main difference between APS and CIC is that APS uses successor features while CIC uses a contrastive estimator for the discriminator. For further details regarding baselines we refer the reader to Appendix C.

**Evaluation:** We follow an identical evaluation to the 2M pre-training setup in URLB. First, we pre-train each RL agent with the intrinsic rewards for 2M steps. Then, we finetune tune each agent to the downstream task with extrinsic rewards in the data-efficient regime of 100k steps. We use 10 seeds across each downstream task for our method and all the baseline algorithms. For baselines, we benchmark against leading knowledge-based, data-based, and competence-based approaches that have been implemented in URLB. All baselines use the same DDPG optimization algorithm to eliminate confounding factors when comparing algorithms.

To ensure that our evaluation statistics are unbiased we use stratified bootstrap confidence intervals to report aggregate statistics across $M$ runs with $N$ seeds as described in *Rliable* (Agarwal et al., 2021) to report statistics for our main results in Fig. 4. Our primary success metric is the interquartile mean (IQM) and the Optimality Gap (OG). IQM discards the top and bottom 25% of runs and then computes the mean. It is less susceptible to outliers than the mean and was shown to be the most reliable statistic for reporting results for RL experiments in Agarwal et al. (2021). OG measures how far a policy is from optimal (expert) performance. To define expert performance we use the convention in URLB, which is the score achieved by a randomly initialized DDPG after 2M steps of finetuning (20x more steps than our finetuning budget).

## 7 RESULTS

In this section we investigate empirical answers to the following research questions: (Q1) How does CIC adaptation efficiency compare to prior competence-based algorithms and exploration algorithms

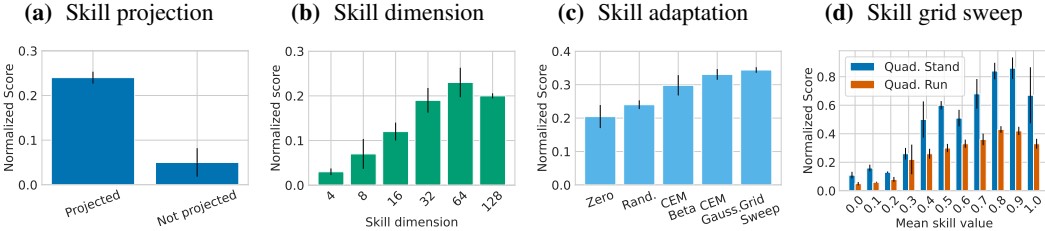

Figure 6: Design choices for pre-training and adapting with skills have significant impact on performance. In (a) and (b) the agent's zero-shot performance is evaluated while sampling skills randomly while in (c) and (d) the agent's performance is evaluated after finetuning the skills vector. *(a)* we show empirically that the projecting skill vectors after sampling them from noise significantly improves the agent's performance. *(b)* The skill dimension is a crucial hyperparameter and, unlike prior methods, CIC scales to large skill vectors achieving optimal performance at 64 dimensional skills. *(c)* We test several adapation strategies and find that a simple grid search performs best given the small 4k step adaptation budget, *(d)* Choosing the right skill vector has substantial impact on performance and grid sweeping allows the agent to select the appropriate skill.

more broadly? (Q2) Qualitatively, does CIC discover structured skills and particularly is it able to do so in environments with high-dimensional state-action spaces? (Q3) Quantitatively, how does CIC behavior compare to prior methods? (Q4) Is skill selection important for efficient adaptation to downstream tasks? (Q5) How does the skill dimension affect the quality of the pre-trained policy?

**Adaptation efficiency of CIC and exploration baslines:** Expert normalized scores of CIC and exploration algorithms from URLB are shown in Fig. 3. We find that CIC substantially outperforms prior competence-based algorithms (DIAYN, SMM, APS) achieving a 91% higher IQM than the next best competence-based method (APS) and, more broadly, achieving a 26% higher IQM than the next best overall baseline (ProtoRL). In further ablations, we find that the contributing factors to CIC's performance are its ability to accommodate substantially larger continuous skill spaces than prior competence-based methods.

**Quantitative analysis of CIC behaviors:** Quantitatively, intrinsic reward profiles during pre-training and behavior entropies are shown in Fig 5. Since the CPC and etnropy terms are on different scale, we pick a default hyparameter of $\alpha = 0.9$ that puts them on equal footing. We find that the CPC intrinsic reward increases through training while the entropy term decreases to a non-zero value. This is what we would expect to see as the discriminator distills behaviors into a set of skills with lower than pure entropy maximization without skill learning. Using a particle estimator for entropy, we find that CIC behavioral entropy is less than APT and greater than DIAYN or CIC without the entropy term. This suggests that the CIC agent has learned non-static skills while DIAYN skills are mostly static. Finally, we find that CIC with $\alpha = 0.0$ (no entropy) on Jaco is optimal for downstream task performance. This is most likely because, unlike Walker and Quadruped, which require locomotion, Jaco tasks require reaching a certain position, so low-entropy skills that take the end effector to a certain end position are favorable to skills that result in periodic motion.

For a qualitative analysis, we refer the reader to Fig. 2 and Appendix J.

**Skill architecture and adaptation ablations:** We find that projecting the skill to a latent space before inputting it as the key for the contrastive loss is an important design decision (see Fig. 6a), most likely because this reduces the diversity of the skill vector making the discriminator task simpler.

We also find empirically that the skill dimension is an important hyperparameter and that larger skills results in better zero-shot performance (see Fig. 6b), which empirically supports the hypothesis posed in Section 3 and Appendix I that larger skill spaces are important for internalizing diverse behaviors. Interestingly, CIC zero-shot performance is poor in lower skill dimensions (e.g. $\dim(z) < 10$), suggesting that when $\dim(z)$ is small CIC likely performs no better than prior competence-based methods such as DIAYN, and that scaling to larger skill dimensions enables CIC to pre-train effectively.

To measure the effect of skill finetuning described in Section 5, we sweep mean skill values along the interval of the uniform prior $[0, 1]$ with a budget of 4k total environment interactions and read out the performance on the downstream task. By sweeping, we mean simply iterating over the interval $[0, 1]$ with fixed step size (e.g. $v = 0, 0.1, \ldots, 0.9, 1$) and setting $z_i = v$ for all $i$. This is not an optimal skill sampling strategy but works well due to the extremely limited number of samples for skill selection.

We evaluate this ablation on the Quadruped Stand and Run downstream tasks. The results shown in Fig. 6 indicate that skill selection affects downstream task performance. The most optimal skill results in $8\times$ better performance than the least optimal skill on Quadruped stand. Ablating the skill dimension, we evaluate the zero-shot performance of the agent in Walker Walk with a fixed skill of $0$ [4] and find that the zero-shot performance monotonically increases from skill dimension $4$ until reaching dimension $64$ and starts decreasing for even higher skill dimensions.

## 8 RELATED WORK

**Supervised Reinforcement Learning:** To date, most of RL research has focused on supervised RL where training is supervised with an extrinsic reward function. Supervised RL has seen many breakthroughs over the last five years (Mnih et al., 2015; Silver et al., 2017; Vinyals et al., 2019; Silver et al., 2018; Berner et al., 2019; Andrychowicz et al., 2020; Schulman et al., 2016; 2017). The field has also produced several stable RL optimization algorithms that have helped accelerated research Haarnoja et al. (2018); Hessel et al. (2018); Lillicrap et al. (2016); Schulman et al. (2017).

**Unsupervised Reinforcement Learning:** The sub-field of unsupervised RL consists of two primary research areas - *unsupervised behavioral learning* and *unsupervised representation learning* (Srinivas & Abbeel, 2021). Unsupervised behavioral learning consists of learning behaviors and exploring the environment without extrinsic rewards. Unsupervised representation learning consists of learning representations without supervision from high-dimensional data such as pixel observations. We evaluate our method on the recently introduced Unsupervised RL Benchmark (URLB) (Laskin et al., 2021), and focus solely on the behavioral aspect of unsupervised RL in order to isolate the core issue preventing prior unsupervised skill discovery methods from exploring effectively on URLB.

**Unsupervised Behavioral Learning:** The aim of unsupervised behavioral learning is to produce diverse behaviors that explore the environment without interacting with an extrinsic reward. Often referred to as intrinsic motivation (Oudeyer et al., 2007), this is typically achieved by defining an intrinsic reward through a self-supervised task. Most behavioral learning algorithms fall into three categories – knowledge-based (Pathak et al., 2017; 2019; Burda et al., 2019b;a) where the agent maximizes the error or uncertainty of some predictive model, data-based (Campos et al., 2021b; Liu & Abbeel, 2021a;b; Mutti et al., 2021; Seo et al., 2021; Yarats et al., 2021b) where the agent maximizes data diversity, and competence-based Eysenbach et al. (2019); Hansen et al. (2020); Liu & Abbeel (2021b); Sharma et al. (2020) where the agent maximizes the mutual information between observable variables and a latent skill vector. We discuss the differences between CIC and the most closely related competence-based exploration algorithms in Appendix D.

**Unsupervised Representation Learning:** Much progress in unsupervised representation learning for RL has been spurred by unsupervised learning in computer vision (Chen et al., 2020; He et al., 2020; Hénaff et al., 2020; Kingma & Welling, 2013) and language (Brown et al., 2020; Devlin et al., 2019; Radford et al., 2019). In RL, the most common approach for representation learning has been by adding it as an auxiliary loss in the supervised RL setting (Jaderberg et al., 2017). More recently, a number of works have investiagted representaton learning with autoencoders (Yarats et al., 2019; Hafner et al., 2019; 2020), siamese networks (Schwarzer et al., 2021a;b; Laskin et al., 2020b; Stooke et al., 2021; Yarats et al., 2021b), and data augmentation (Laskin et al., 2020a; Yarats et al., 2021a;c).

## 9 CONCLUSION

We have introduced a new competence-based algorithm – Contrastive Intrinsic Control (CIC) – which enables more effective exploration than prior unsupervised skill discovery algorithms by explicitly encouraging diverse behavior while distilling predictable behaviors into skills with a contrastive discriminator. We showed that CIC is the first competence-based approach to achieve leading performance on URLB. We hope that this encourages further research in unsupervised skill discovery toward building more powerful exploration agents.

---

[4]The performance will, of course, be much better if we finetune the skill, but we cannot do so for zero-shot evaluation.

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

## A    COMPETENCE-BASED EXPLORATION ALGORITHMS

The competence-based algorithms considered in this work aim to maximize $I(\tau; s)$. The algorithms differ by ho they decompose mutual information, whether they explicitly maximize behavioral entropy, their skill space (discrete or continuous) and their intrinsic reward structure. We provide a list of common competence-based algorithms in Table 1.

Table 1: Competence-based Unsupervised Skill Discovery Algorithms

| Algorithm | Intrinsic Reward | Decomposition | Explicit $\max \mathcal{H}(\tau)$ | Skill Dim. | Skill Space |
|---|---|---|---|---|---|
| SSN4HRL (Florensa et al., 2018) | $\log q_\psi(z\|s_t)$ | $H(z) - H(z\|\tau)$ | No | 6 | discrete |
| VIC (Gregor et al., 2017) | $\log q_\psi(z\|s_H))$ | $H(z) - H(z\|\tau)$ | No | 60 | discrete |
| VALOR (Achiam et al., 2018) | $\log q_\psi(z\|s_{1:H})$ | $H(z) - H(z\|\tau)$ | No | 64 | discrete |
| DIAYN (Eysenbach et al., 2019) | $\log q_\psi(z\|s_t)$ | $H(z) - H(z\|\tau)$ | No | 128 | discrete |
| DADS (Sharma et al., 2020) | $q_\psi(s'\|z,s) - \sum_i \log q(s'\|z_i, s)$ | $H(\tau) - H(\tau\|z)$ | Yes | 5 | continuous |
| VISR (Hansen et al., 2020) | $\log q_\psi(z\|s_t)$ | $H(z) - H(z\|\tau)$ | No | 10 | continuous |
| APS (Liu & Abbeel, 2021b) | $F_{\text{Successor}}(s\|z) + \mathcal{H}_{\text{particle}}(s)$ | $\mathcal{H}(\tau) - \mathcal{H}(\tau\|z)$ | Yes | 10 | continuous |
| CIC (Ours) | $F_{\text{CPC}}(s, s'\|z) + \mathcal{H}_{\text{particle}}(s, s')$ | $\mathcal{H}(\tau) - \mathcal{H}(\tau\|z)$ | Yes | 64 | continuous |

Table 2: A list of competence-based algorithms. We describe the intrinsic reward optimized by each method and the decomposition of the mutual information utilized by the method. We also note whether the method explicitly maximizes state transition entropy. Finally, we note the maximal dimension used in each work and whether the skills are discrete or continuous. All methods prior to CIC only support small skill spaces, either because they are discrete or continuous but low-dimensional.

## B    DEEP DETERMINISTIC POLICY GRADIENT (DDPG)

A DDPG is an actor-critic RL algorithm that performs off-policy gradient updates and learns a Q function $Q_\phi(s, a)$ and an actor $\pi_\theta(a|s)$. The critic is trained by satisfying the Bellman equation.

$$\mathcal{L}_Q(\phi, \mathcal{D}) = \mathbb{E}_{(s_t, a_t, r_t, s_{t+1}) \sim \mathcal{D}} \left[ \left( Q_\phi(s_t, a_t) - r_t - \gamma Q_{\bar{\phi}}(s_{t+1}, \pi_\theta(s_{t+1})) \right)^2 \right]. \tag{7}$$

Here, $\bar{\phi}$ is the Polyak average of the parameters $\phi$. As the critic minimizes the Bellman error, the actor maximizes the action-value function.

$$\mathcal{L}_\pi(\theta, \mathcal{D}) = \mathbb{E}_{s_t \sim \mathcal{D}} \left[ Q_\phi(s_t, \pi_\theta(s_t)) \right]. \tag{8}$$

## C    BASELINES

For baselines, we choose the existing set of benchmarked unsupervised RL algorithms on URLB. We provide a quick summary of each method. For more detailed descriptions of each baseline we refer the reader to URLB (Laskin et al., 2021)

*Competence-based Baselines:* CIC is a competence-based exploration algorithm. For baselines, we compare it to DIAYN (Eysenbach et al., 2019), SMM (Lee et al., 2019), and APS (Liu & Abbeel, 2021b). Each of these algorithms is described in Table 1. Notably, APS is a recent state-of-the-art competence-based method that is the most closely related algorithm to the CIC algorithm. CIC and APS differ in their discriminator.

*Knowledge-based Baselines:* For knowledge-based baselines, we compare to ICM Pathak et al. (2017), Disagreement Pathak et al. (2019), and RND Burda et al. (2019b). ICM and RND train a dynamics model and random network prediction model and define the intrinsic reward to be proportional to the prediction error. Disagreement trains an ensemble of dynamics models and defines the intrinsic reward to be proportional to the uncertainty of an ensemble.

*Data-based Baselines:* For data-based baselines we compare to APT (Liu & Abbeel, 2021a) and ProtoRL (Yarats et al., 2021b). Both methods use a particle estimator to estimate the state visitation entropy. ProtoRL also performs discrete contrastive clustering as in Caron et al. (2020) as an auxiliary task and uses the resulting clusters to compute the particle entropy. While ProtoRL is more effective than APT when learning from pixels, on state-based URLB APT is competitive with ProtoRL. Our method CIC is effectively a skill-conditioned APT agent with a contrastive discriminator.

## D  RELATION TO PRIOR SKILL DISCOVERY METHODS

The most closely relatd prior algorithm to CIC is APS Liu & Abbeel (2021b). Both CIC and APS use the $\mathcal{H}(\tau) - \mathcal{H}(\tau|z)$ decomposition of the mutual information and both used a particle estimator (Singh et al., 2003) to compute the state entropy as in Liu & Abbeel (2021a). The main difference between CIC and APS is the discriminator. APS uses successor features as in Hansen et al. (2020) for its discriminator while CIC uses a noise contrastive estimator. Unlike successor features, which empirically only accommodate low-dimensional continuous skill spaces (see Table 1), the noise contrastive discriminator is able to leverage higher continuous dimensional skill vectors.

The CIC discriminator is similar to the one used in DISCERN (Warde-Farley et al., 2018), a goal-condition unsupervised RL algorithm. Both methods use a contrastive discriminator by sampling negatives and computing an inner product between queries and keys. The main differences are (i) that DISCERN maximizes $I(\tau; g)$ where $g$ are image goal embeddings while CIC maximizes $I(\tau; z)$ where $z$ are abstract skill vectors; (ii) DISCERN uses the DIAYN-style decomposition $I(\tau; g) = H(g) - H(g|\tau)$ while CIC decomposes through $H(\tau) - H(\tau|z)$, and (iii) DISCERN discards the $H(g)$ term by sampling goals uniformly while CIC explicitly maximizes $\mathcal{H}(\tau)$. While DISCERN and CIC share similarities, DISCERN operates over image goals while CIC operates over abstrac skill vectors so the two methods are not directly comparable.

Finally, another similar algorithm to CIC is DADS (Sharma et al., 2020) which also decomposes through $H(\tau) - H(\tau|z)$. While CIC uses a contrastive density estimate for the discriminator, DADS uses a maximum likelihood estimator similar to DIAYN. DADS maximizes $I(s'|s, z)$ and estimates entropy $\mathcal{H}(s'|s)$ by marginalizing over $z$ such that $\mathcal{H}(s'|s) = -\log \sum_i q(s'|s, z_i)$ while CIC uses a particle estimator. Interestingly, the DADS intrinsic reward $r_i \propto \log\left(q(s'|s, z) / \sum_i q(s'|s, z_i)\right)$ looks similar to the CIC objective with zero entropy, since marginalizing over $z$ to compute entropy is similar to sampling negatives for a contrastive discriminator.

# E    FULL CIC ALGORITHM

The full CIC algorithm with both pre-training and fine-tuning phases is shown in Algorithm 1. We pre-train CIC for 2M steps, and finetune it on each task for 100k steps.

---

**Algorithm 1** Contrastive Intrinsic Control

---

**Require:** Initialize all networks: encoders $g_{\psi_1}$ and $g_{\psi_2}$, actor $\pi_\theta$, critic $Q_\phi$, replay buffer $\mathcal{D}$.
**Require:** Environment (env), $M$ downstream tasks $T_k, k \in [1, \ldots, M]$.
**Require:** pre-train $N_{\mathrm{PT}} = 2M$ and fine-tune $N_{\mathrm{FT}} = 100K$ steps.

1: **for** $t = 1..N_{\mathrm{PT}}$ **do**                                                ▷ Part 1: Unsupervised Pre-training
2:       Sample and encode skill $z \sim p(z)$ and $z \leftarrow g_{\psi_2}(z)$
3:       Encode state $s_t \leftarrow g_{\psi_1}(s_t)$ and sample action $a_t \leftarrow \pi_\theta(s_t, z) + \epsilon$ where $\epsilon \sim \mathcal{N}(0, \sigma^2)$
4:       Observe next state $s_{t+1} \sim P(\cdot|s_t, a_t)$
5:       Add transition to replay buffer $\mathcal{D} \leftarrow \mathcal{D} \cup (s_t, a_t, s_{t+1})$
6:       Sample a minibatch from $\mathcal{D}$, compute contrastive loss in Eq.3 and update encoders $g_{\psi_1}, g_{\psi_2}$, compute CIC intrinsic reward with Eq. 6 and update actor $\pi_\theta$ and critic $Q_\phi$
7: **end for**
8: **for** $T_k \in [T_1, \ldots, T_M]$ **do**                                          ▷ Part 2: Supervised Fine-tuning
9:       Initialize all networks with weights from pre-training phase and an empty replay buffer $\mathcal{D}$.
10:       **for** $t = 1 \ldots 4,000$ **do**
11:            Take random action $a_t \sim \mathcal{N}(0, 1)$
12:            Select skill with grid sweep over unit interval $[0, 1]$ every 100 steps
13:            Sample minibatch from $\mathcal{D}$ and update actor $\pi_\theta$ and critic $Q_\phi$
14:       **end for**
15:       Fix skill $z$ that achieved highest extrinsic reward during grid sweep.
16:       **for** $t = 4,000 \ldots N_{\mathrm{FT}}$ **do**
17:            Encode state $s_t \leftarrow g_{\psi_1}(s_t)$ and sample action $a_t \leftarrow \pi_\theta(s_t, z) + \epsilon$ where $\epsilon \sim \mathcal{N}(0, \sigma^2)$
18:            Observe next state and reward $s_{t+1}, r_t^{\mathrm{ext}} \sim P(\cdot|s_t, a_t)$
19:            Add transition to replay buffer $\mathcal{D} \leftarrow \mathcal{D} \cup (s_t, a_t, r_t^{\mathrm{ext}}, s_{t+1})$
20:            Sample minibatch from $\mathcal{D}$ and update actor $\pi_\theta$ and critic $Q_\phi$.
21:       **end for**
22:       Evaluate performance of RL agent on task $T_k$
23: **end for**

---

## F  HYPER-PARAMETERS

Baseline hyperparameters are taken from URLB Laskin et al. (2021), which were selected by performing a grid sweep over tasks and picking the best performing set of hyperparameters. Similarly, we also performed a grid sweep for CIC to pick the best performing set of hyperparameters. All hyperparameters are the same across all domains except for $\alpha$ which is set to $\alpha = 0.9$ for Walker and Quadruped domains. Note that $\alpha = 0.9$ results in equal weighing of the CPC and particle entropy terms since their absolute values are on different scales. For Jaco, we found $\alpha = 0.0$ to work best, which means that only the discriminator contributes to the intrinsic reward. We hypothesize that particle entropy maximization is not important for Jaco arm because it is fixed and has no way of falling over like Walker and Quadruped, such that meaningful behaviors can be learned with the discriminator alone.

Table 3: Hyper-parameters used for CIC .

| DDPG hyper-parameter | Value |
| --- | --- |
| Replay buffer capacity | $10^6$ |
| Action repeat | 1 states-based and 2 for pixels-based |
| Seed frames | 4000 |
| $n$-step returns | 3 |
| Mini-batch size | 1024 states-based and 256 for pixels-based |
| Seed frames | 4000 |
| Discount ($\gamma$) | 0.99 |
| Optimizer | Adam |
| Learning rate | $10^{-4}$ |
| Agent update frequency | 2 |
| Critic target EMA rate ($\tau_Q$) | 0.01 |
| Features dim. | 1024 states-based and 50 for pixels-based |
| Hidden dim. | 1024 |
| Exploration stddev clip | 0.3 |
| Exploration stddev value | 0.2 |
| Number pre-training frames | up to $2 \times 10^6$ |
| Number fine-turning frames | $1 \times 10^5$ |
| CIC hyper-parameter | Value |
| Skill dim | 64 continuous |
| Prior | Uniform [0,1] |
| $\alpha$ | 0.9 Walker, Quadruped, 0.0 Jaco |
| Skill sampling frequency (steps) | 50 |
| State net arch. $g_{\psi_1}(s)$ | $\dim(\mathcal{O}) \to 1024 \to 1024 \to 64$ ReLU MLP |
| Skill net arch. $g_{\psi_2}(z)$ | $64 \to 1024 \to 1024 \to 64$ ReLU MLP |
| Prediction net arch. | $64 \to 1024 \to 1024 \to 64$ ReLU MLP |

## G   RAW NUMERICAL RESULTS

We provide a list of raw numerical results for finetuning CIC and baselines in Table 4.

| Domain | Task | Pre-trainining for $2 \times 10^6$ environment steps |  |  |  |  |  |  |  |  |  |
|---|---|---|---|---|---|---|---|---|---|---|---|
|  |  | DDPG | CIC | ICM | Disagreement | RND | APT | ProtoRL | SMM | DIAYN | APS |
| Walker | Flip | 538±27 | **671 ± 34** | 417±16 | 346±13 | 474±39 | 544±14 | 456±12 | 450±24 | 319±17 | 465±20 |
|  | Run | 325±25 | **421 ± 39** | 247±21 | 208±15 | 406±30 | 392±26 | 306±13 | **426±26** | 158±8 | 134±16 |
|  | Stand | 899±23 | **947 ± 5** | 859±23 | 746±34 | 911±5 | **942±6** | 917±27 | 924±12 | 695±46 | 721±44 |
|  | Walk | 748±47 | **895 ± 24** | 627±42 | 549±37 | 704±30 | 773±70 | 792±41 | 770±44 | 498±27 | 527±79 |
| Quadruped | Jump | 236±48 | **684 ± 23** | 178±35 | 389±62 | 637±12 | 648±18 | 617±44 | 96±7 | **660±43** | 463±51 |
|  | Run | 157±31 | 424 ± 29 | 110±18 | 337±30 | 459±6 | **492±14** | 373±33 | 96±6 | 433±29 | 281±17 |
|  | Stand | 392±73 | 789 ± 45 | 312±68 | 512±89 | 766±43 | 872±23 | 716±56 | 123±11 | **851±43** | 542±53 |
|  | Walk | 229±57 | 673 ± 68 | 126±27 | 293±37 | 536±39 | **770±47** | 412±54 | 80±6 | 576±81 | 436±79 |
| Jaco | Reach bottom left | 72±22 | **127 ± 15** | 111±11 | **124±7** | 110±5 | 103±8 | **129±8** | 45±7 | 39±6 | 76±8 |
|  | Reach bottom right | 117±18 | **172 ± 9** | 97±9 | 115±10 | 117±7 | 100±6 | 132±8 | 46±11 | 38±5 | 88±11 |
|  | Reach top left | 116±22 | **156 ± 21** | 82±14 | 106±12 | 99±6 | 73±12 | 123±9 | 36±3 | 19±4 | 68±6 |
|  | Reach top right | 94±18 | **191 ± 5** | 103±11 | 139±7 | 100±6 | 90±10 | 159±7 | 47±6 | 28±6 | 76±10 |

Table 4: Performance of CIC and baselines on state-based URLB after first pre-training for $2 \times 10^6$ steps and then finetuning with extrinsic rewards for $1 \times 10^5$.

## H   LEARNING CURVES FOR DOWNSTREAM ADAPTATION PHASE

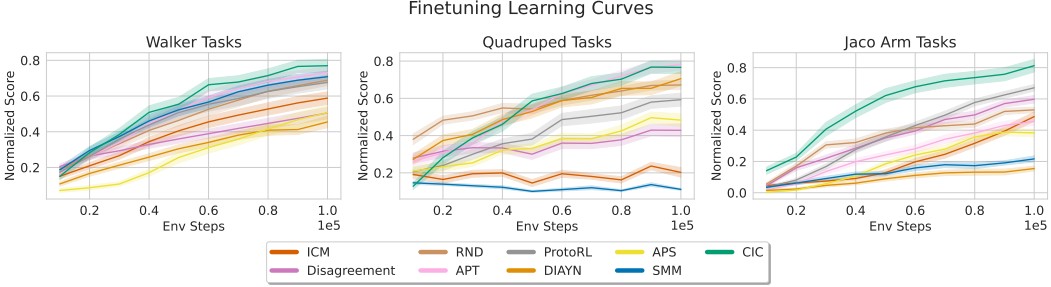

Figure 7: Learning curves for finetuning pre-trained agents for 100k steps. Task performance is aggregated for each domain, such that each curve represents the mean normalized scores over $4 \times 10 = 40$ seeds. The shaded regions represent the standard error. CIC surpasses the performance of the prior state-of-the-art on Walker and Jaco tasks while tying on Quadruped. CIC is the only algorithm that performs consistently well across all three domains.

## I   TOY EXAMPLE TO ILLUSTRATE THE NEED FOR LARGER SKILL SPACES

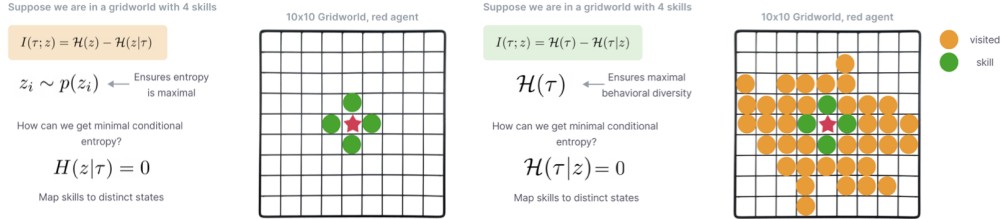

Figure 8: A gridworld example motivating the need for large skill spaces. In this environment, we place an agent in a $10 \times 10$ gridworld and provide the agent access to four discrete skills. We show that the mutual information objective can be maximized by mapping these four skills to the nearest neighboring states resulting in low behavioral diversity and exploring only four of the hundred available states.

We illustrate the need for larger skill spaces with a gridworld example. Suppose we have an agent in a $10 \times 10$ sized gridworld and that we have four discrete skills at our disposal. Now let $\tau = s$ and consider how we may achieve maximal $I(\tau; z)$ in this setting. If we decompose $I(\tau; z) = \mathcal{H}(z) - \mathcal{H}(z|\tau)$ then we can achieve maximal $\mathcal{H}(z)$ by sampling the four skills uniformly $z \sim p(z)$. We can achieve $\mathcal{H}(z|\tau) = 0$ by mapping each skill to a distinct neighboring state of the agent. Thus, our mutual information is maximized but as a result the agent only explores four out of the hundrend available states in the gridworld.

Now suppose we consider the second decomposition $I(\tau; z) = \mathcal{H}(\tau) - \mathcal{H}(\tau|z)$. Since the agent is maximizing $\mathcal{H}(\tau)$ it is likely to visit a diverse set of states at first. However, as soon as it learns an accurate discriminator we will have $\mathcal{H}(\tau|z)$ and again the skills can be mapped to neighboring states to achieve minimal conditional entropy. As a result, the skill conditioned policy will only be able to reach four out of the hundrend possible states in this gridworld. This argument is shown visually in Fig. 8.

Skill spaces that are too large can also be an issue. Consider if we had 100 skills at our disposal in the same gridworld. Then the agent could minimize the conditional entropy by mapping each skill to a unique state which would result in the agent memorizing the environment by finding a one-to-one mapping between states and skills. While this is a potential issue it has not been encountered in practice yet since current competence-based methods support small skill spaces relative to the observation space of the environment.

## J    QUALITATIVE ANALYSIS OF SKILLS

We provide two additional qualitative analyses of behaviors learned with the CIC algorithm. First, we take a simple pointmass setting and set the skill dimension to 1 in order to ablate the skills learned by the CIC agent in a simple setting. We sweep over different values of $z$ and plot the behavioral flow vector field (direction in which point mass moves) in Fig.9. We find that the pointmass learns skills that produce continuous motion and that the direction of the motion changes as a function of the skill value. Near the origin the pointmass learns skills that span all directions, while near the edges the point mass learns to avoid wall collisions. Qualitatively, many behaviors are periodic.

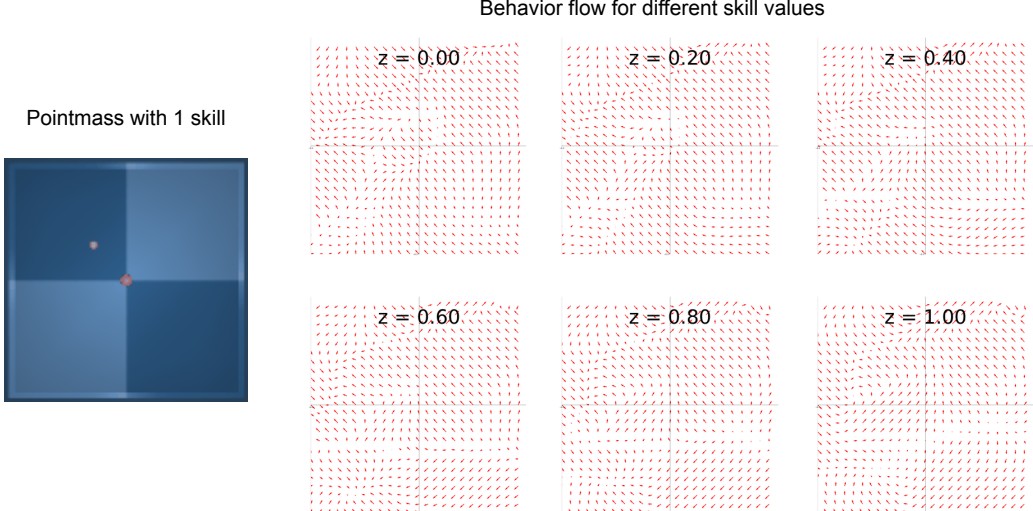

Figure 9: Learning curves for finetuning pre-trained agents for 100k steps. Task performance is aggregated for each domain, such that each curve represents the mean normalized scores over $4 \times 10 = 40$ seeds. The shaded regions represent the standard error. CIC surpasses the performance of the prior state-of-the-art on Walker and Jaco tasks while tying on Quadruped. CIC is the only algorithm that performs consistently well across all three domains.

Qualitatively, we find that methods like DIAYN that only support low dimensional skill vectors and do not explicitly incentivize diverse behaviors in their objective produce policies that map skills to a small set of static behaviors. These behaviors shown in Fig. 10 are non-trivial but also have low behavioral diversity and are not particularly useful for solving the downstream task. This observation is consistent with Zahavy et al. (2021) where the authors found that DIAYN maps to static "yoga" poses in DeepMind Control. In contrast, behaviors produce by CIC are dynamic resulting flipping, jumping, and locomotive behaviors that can then be adapted to efficiently solve downstream tasks.

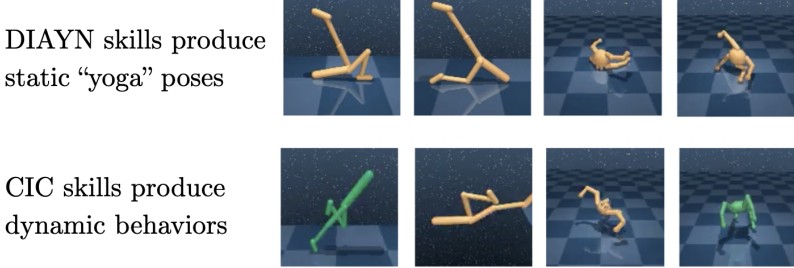

Figure 10: Qualitative visualization of DIAYN and CIC pre-training on the Walker and Quadruped domains from URLB. Confirming findings in prior work Zahavy et al. (2021), we also find that DIAYN policies produce static but non-trivial behaviors mapping to "yoga" poses while CIC produces diverse and dynamic behaviors such as walking, flipping, and standing. Though it's hard to see from these images, all the DIAYN skills get stuck in frozen poses while the CIC skills are producing dynamic behavior with constant motion.

## K  OpenAI Gym vs. DeepMind control: How Early Termination Leaks Extrinsic Signal

Prior work on unsupervised skill discovery for continuous control (Eysenbach et al., 2019; Sharma et al., 2020) was evaluated on OpenAI Gym (Brockman et al., 2016) and showed diverse exploration on Gym environments. However, Gym environment episodes terminate early when the agent loses balance, thereby leaking information about the extrinsic task (e.g. balancing or moving). However, DeepMind Control (DMC) episodes have a fixed length of 1k steps. In DMC, exploration is therefore harder since the agent needs to learn to balance without any extrinsic signal.

To evaluate whether the difference in the two environments has impact on competence-based exploration, we run DIAYN on the hopper environments from both Gym and DMC. We compare to ICM, a popular exploration baseline, and a Fixed baseline where the agent receives an intrinsic reward of 1 for each timestep and no algorithms receive extrinsic rewards. We then measure the extrinsic reward, which loosely corresponds to the diversity of behaviors learned. Our results in Fig. 11 show that indeed DIAYN is able to learn diverse behaviors in Gym but not in DMC while ICM is able to learn diverse behaviors in both environments. Interestingly, the Fixed baseline achieves the highest reward on the Gym environment by learning to stand and balance. These results further motivate us to evaluate on URLB which is built on top of DMC.

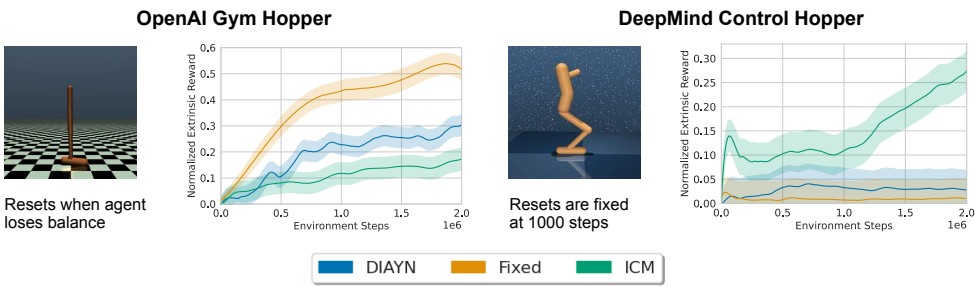

Figure 11: To empirically demonstrate issues inherent to competence-based exploration methods, we run DIAYN (Eysenbach et al., 2019) and compare it to ICM (Pathak et al., 2017) and a *Fixed* baseline where the agent receives an intrinsic reward of 1.0 for each timestep and no extrinsic reward on both OpenAI Gym *(episode resets when agent loses balance)* and DeepMind Control (DMC) *(episode is fixed for 1k steps)* Hopper environments. Since Gym and DMC rewards are on different scales, we normalize rewards based on the maximum reward achieved by any algorithm ( 1k for Gym,  3 for DMC). While DIAYN is able to achieve higher extrinsic rewards than ICM on Gym, the Fixed intrinsic reward baseline performs best. However, on DMC the Fixed and DIAYN agents achieve near-zero reward while ICM does not. This is consistent with findings of prior work that DIAYN is able to learn diverse behaviors in Gym (Eysenbach et al., 2019) as well as the observation that DIAYN performs poorly on DMC environments (Laskin et al., 2021)

## L PSEUDOCODE FOR THE CONTRASTIVE DISCRIMINATOR IN CIC

CIC consists of two terms $I(\tau; z) = H(\tau) - H(\tau|z) \geq H(\tau) + \mathbb{E}[\log q(\tau|z)]$ the entropy $H(\tau)$ is estimated with a particle estimator Singh et al. (2003); Liu & Abbeel (2021a) while the discriminator $q(\tau|z)$ is estimated with a contrastive loss introduced in this work. Note that contrastive learning in CIC is different than prior vision-based contrastive learning such as CURL Laskin et al. (2020b), since we are not performing contrastive learning over augmented images but rather over state transitions and skills. The contrastive objective in CIC is used for unsupervised learning of behaviors while in CURL it is used for unsupervised learning of visual features.

We provide pseudocode for discriminator below:

```
def discriminator_loss(states, next_states, skills, temp):
    """
    - states and skills are sampled from replay buffer
    - skills were sampled from uniform dist [0,1] during agent rollout
    - states / next_states: dim (B, D_state)
    - skills: dim (B, D_skill)
    """

    transitions = concat(states, next_states, dim=1)

    query = skill_net(skills) # (B, D_hidden) -> (B, D_hidden)
    key = transition_net(transitions) # (B, 2*D_state) -> (B, D_hidden)

    query = normalize(query, dim=1)
    key = normalize(key, dim=1)

    logits = matmul(query, key.T) / temp # (B, B)

    # positives are on diagonal, negatives are off diagonal
    # for each skill, negatives are sampled from transitions
    # while skills are fixed
    loss = cross_entropy(logits)

    return loss
```

Listing 1: CIC discriminator loss

We note that this is substantially different from prior contrastive learning works in RL such as CURL (Laskin et al., 2020b), which perform contrastive learning over images.

```
def curl_loss(obs, W, temp):
    """
    - observation images are sampled from replay buffer
    - obs: dim (B, C, H, W)
    - W: projection matrix (D_hidden, D_hidden)
    """

    query = aug(obs)
    key = aug(obs)

    query = cnn_net(query) # (B, D_hidden)
    key = cnn_net(key) # (B, D_hidden)

    logits = matmul(matmul(query, W), key.T) / temp # (B, B)

    # positives are on diagonal
    # negatives are off diagonal
    loss = cross_entropy(logits)

    return loss
```

Listing 2: CURL contrastive loss

## M    ON TIGHTER ESTIMATES OF MUTUAL INFORMATION

In this work we have presented CIC - a new competence-based algorithm that achieves leading performance on URLB compared to prior unsupervised RL methods. We've shown that CIC results in a tighter lower bound on mutual information than CPC by including the entropy term.

One might wonder whether estimating the exact mutual information (MI) or maximizing the tightest lower bound thereof is really the goal for unsupervised RL. In unsupervised representation learning, state-of-the-art methods like CPC and SimCLR maximize the lower bound of MI based on Noise Contrastive Estimation (NCE). However, as proven in CPC (Oord et al., 2018) and illustrated in Poole et al. (2019) NCE is upper bounded by $\log N$, meaning that the bound is loose when the MI is larger than $\log N$. Nevertheless, these methods have been repeatedly shown to excel in practice. In Tschannen et al. (2020) the authors show that the effectiveness of NCE results from the inductive bias in both the choice of feature extractor architectures and the parameterization of the employed MI estimators.

We have a similar belief for unsupervised RL - that with the right parameterization and inductive bias, the MI objective will facilitate behavior learning in unsupervised RL. This is why CIC lower bounds MI with (i) the particle based entropy estimator to ensure explicit exploration and (ii) a contrastive conditional entropy estimator to leverage the power of contrastive learning to discriminate skills. As demonstrated in our experiments, CIC outperforms prior methods, showing the effectiveness of optimizing an intrinsic reward with the CIC MI estimator.

## N    LIMITATIONS

While CIC achieves leading results on URLB, we would also like to address its limitations. First, in this paper we only consider MDPs (and not partially observed MDPs) where the full state is observable. We focus on MDPs because generating diverse behaviors in environments with large state spaces has been the primary bottleneck for competence-based exploration. Combining CIC with visual representation learning to scale this method to pixel-based inputs is a promising future direction for research not considered in this work. Another limitation is that our adaptation strategy to downstream tasks requires finetuning. Since we learn skills, it would be interesting to investigate alternate ways of adapting that would enable zero-shot generalization such as learning generalized reward functions during pre-training.

