# OpenReview forum: "CIC: Contrastive Intrinsic Control for Unsupervised Skill Discovery"
_ICLR.cc/2022/Conference — ICLR 2022 Submitted_

### Official Review · Reviewer_VTZ1 · 2021-10-22

**Correctness:** 4
**Technical Novelty And Significance:** 2
**Empirical Novelty And Significance:** 3
**Recommendation:** 6
**Confidence:** 4

**Main Review:**

I report below some detailed comments and concerns that the authors might address in their author response.

MUTUAL INFORMATION OBJECTIVE

1) There is now a bunch of works targeting the mutual information between a code and the visited states to the purpose of unsupervised RL (some of them are summarized in Table 1 of the paper). Every work is proposing a lower bound to the mutual information. I was wondering if there is a formal way to compare the lower bound of CIC with previous works. Which one is the tightest? Do the authors believe that getting the tightest bound of the MI is really the goal in this setting, or even maximizing the exact MI would not be necessarily better than other methods (i.e., a good inductive bias matters more than the MI approximation)?

2) Since the work motivates the approach as an approximation of the MI, as it is common in previous works as well, I was wondering if the authors also considered a direct non-parametric estimation of the MI (e.g., https://journals.aps.org/pre/pdf/10.1103/PhysRevE.69.066138) instead of independent estimates of the state entropy and the conditional entropy. Especially, such a direct estimation would not require the additional hyper-parameter $\alpha$.

METHODOLOGY

3) Can the authors clarify what is the entropy term $H(\tau)$ denoting? Is it the entropy of state-to-state transitions, or the entropy of the joint probability of two states within a trajectory, or something else? Especially, what is the intuition behind using $\tau$ instead of $s$?

4) The method is built upon a non-specific base algorithm (DDPG) that was originally developed for standard RL, i.e. RL problems where the reward does not change over time but it comes from a consistent reward function. Do the authors experienced any instability working with the non-Markovian intrinsic rewards? Do they believe that the methodology could benefit from an objective-specific algorithm, i.e., an algorithm carefully designed to work with this kind of intrinsic reward?

EXPERIMENTS

5) How can we rule out the possibility that CIC is just a better way to pre-train a DDPG agent in these settings w.r.t. other baselines? This would be significant anyway, but do the authors believe that the same results would generalize to different base algorithms (say TRPO, SAC, A2C...)?

6) Moreover, DDPG is known to be quite strong on continuous control tasks. Do the authors believe that the combination of DDPG and CIC would be successful in different settings (e.g., visual discrete domains such as Atari games) as well? Or perhaps the base algorithm should be selected to accomodate the specific domain?

7) CIC is quite similar to APS (Liu and Abbeel, 2021), as they both employ non-parametric entropy estimation and a discriminator loss, based on contrastive predictive coding and successor representations respectively. However, in the reported results CIC is way better than APS. Do the authors think it is the different discriminator loss the main cause for this performance gap, or there is some other factor at play? Can they confront the discriminator rewards of APS and CIC (as in Figure 5)?

8) From my understanding, the empirical results are not directly comparable with the URLB (Laskin et al., 2021) despite a very similar setting. I see that the benchmark is very recent, and thus should be considered concurrent to this work, but I believe that reporting a direct comparison with their results would further strengthen the empirical analysis.

9) The results section seems to imply that a key factor under the improvement over previous competence-based methods is the ability of CIC to cope with a larger skill space. Can the authors clarify why previous methods are prevented to work with a comparable skill space? Can they also provide a comparison with previous methods when working with the same (potentially lower-dimensional) skill space?

MINOR
- Adaptation efficiency paragraph: Fig. 3 is reported instead of Fig. 4;
- The normalized score of Fig. 6 does not seem to match the one of Fig. 4;
- It is not easy to track the different baselines in Fig. 4 (top). What is this plot representing?

**Summary Of The Paper:**

This paper tackles the problem of unsupervised pre-training of a (code-conditioned) policy to improve the performance of downstream RL tasks. In line with previous works in unsupervised skills discovery, it proposes a method, called CIC, to maximize a variational lower bound to the mutual information between the code and the visited states. The lower bound is obtained through the combination of non-parametric state entropy estimation and a contrastive predictive coding loss for the conditional entropy. The paper provides an empirical analysis of CIC over a set of continuous control domains.

**Summary Of The Review:**

To my understanding, the main selling points of this paper are:
- It tackles the very relevant problem of unsupervised pre-training for reinforcement learning;
- The methodology is clear, and a quite natural extension of previous works in unsupervised skills discovery literature;
- Strong empirical results, CIC seems to advance significantly the state-of-the-art performance of unsupervised pre-training in continuous control domains.

Instead, potential shortcomings are:
- The novelty seems limited, as CIC is essentially similar to APS (Liu and Abbeel, 2021) with a different discriminator loss (which has been employed for unsupervised skills discovery before);
- It is not completely clear from the paper what are the specific factors that lead to such a performance improvement over previous works.

Whereas the reported empirical progress might be a sufficient reason for acceptance, my current evaluation is just slightly positive in consideration of the mentioned concerns. I do not think the limited novelty is a crucial problem here, if the authors could better clarify in their response how the CIC methodology is so successful, I will consider raising my score to a clear accept.

---

> ### Author Response · Authors · 2021-11-14
> **[Part 1/3] Thank you for your thorough review and valuable feedback! We address all questions raised by the reviewer.**
>
> Thank you for the thorough review and bringing up points for clarification! We’re glad you found the results compelling and the problem to be relevant to the community. We address all questions and clarifications below:
>
> **Q1:** *Do the authors believe that getting the tightest bound of the MI is really the goal in this setting, or even maximizing the exact MI?*
>
> It’s a great question whether estimating the exact MI or maximizing the tightest bound of MI is really the goal for unsupervised RL! In unsupervised representation learning, state-of-the-art methods like CPC and SimCLR maximize the lower bound of MI based on noise contrastive estimation (NCE). However, as proven in CPC and illustrated in Poole et al 2019, NCE is upper bounded by the logarithm of the sample size, meaning that the bound is loose when the MI is larger than the logarithm of the sample size. Nevertheless, these methods have been repeatedly shown to excel in practice. In Tschannen et al, 2021, the authors show that the effectiveness of NCE results from the inductive bias in both the choice of feature extractor architectures and the parametrization of the employed MI estimators.
>
> We have a similar belief for unsupervised RL - that with the right parametrization and inductive bias, the MI objective will facilitate behavior learning in unsupervised RL. This is why CIC lower bounds MI with (i) the particle based entropy estimator to ensure explicit exploration (and break the curse of the logarithm of sample size) and (ii) a contrastive conditional entropy estimator to leverage the power of contrastive learning to discriminate skills. As demonstrated in our experiments, CIC outperforms prior methods, showing the effectiveness of optimizing the CIC MI estimator.
>
> On Variational Bounds of Mutual Information, Poole et al, 2019.
>
> On Mutual Information Maximization for Representation Learning, Tschannen et al, 2021.
>
> **Q2:** *I was wondering if the authors also considered a direct non-parametric estimation of the MI?*
>
> This is a great question! The reason we (and prior works) must use a parametric estimation of MI is that we need to backpropagate through $z$ in order to learn a skill space that will produce useful exploratory behaviors. A fully nonparametric estimate of MI would not allow us to backpropagate through $z$. However, in our decomposition $I(\tau;z) = H(\tau) - H(\tau|z)$, we use a non-parametric estimate for $H(\tau)$ since we only need it to define part of the intrinsic reward, so we do not need to backpropagate through it. In this sense the CIC MI estimator is part non-parametric / parametric.
>
> **Q3:** *Is it the entropy of state-to-state transitions, or the entropy of the joint probability of two states within a trajectory, or something else? Especially, what is the intuition behind using $s,s’$ instead of $s$?*
>
> Yes, it is the entropy of state transitions. Because CIC (and most prior competence-based methods) operate in the MDP setting, the state $s$ is sufficient to describe the full state of the environment. The intuition for using the transition $s, s’$ as opposed to the single state $s$ is that prior works (DADS, RVIC) showed that methods that only use the state can lead to the discriminator simply memorizing the terminal state that an agent reaches. This can be undesirable. For example, for behaviors useful in the walker environment, the terminal state of the joint positions / velocities doesn’t matter. Instead, we care about locomotion behaviors which are better described in terms of state transitions.
>
> **Q4:** *Did the authors experience any instability working with the non-Markovian intrinsic rewards? Do they believe that the methodology could benefit from an objective-specific algorithm, i.e., an algorithm carefully designed to work with this kind of intrinsic reward?*
>
> For fully observed MDPs considered in this work, we did not experience any instability issues in either CIC or the baseline algorithms dealing with non-stationary intrinsic rewards. However, the URLB paper did point out that learning with non-stationary intrinsic rewards from pixels leads to instability, which is a very interesting research question but is more related to representation learning for exploration in general rather than the specific algorithm (CIC) considered in this work.
>
> In supervised RL, engineering rewards that are bounded (e.g. $[0,1]$) typically leads to more stable RL optimization. We would expect the same to be true for intrinsic reward specification in unsupervised RL. However, to the best of our knowledge this question of intrinsic reward bounding has not been thoroughly studied in literature and we do not focus on it in this work since it is separate from our main research question - why do competence-based methods perform poorly on URLB? However, algorithms better suited for non-stationary intrinsic rewards would be quite interesting to explore for future work!
>
> [continued]

---

> > ### Author Response · Authors · 2021-11-14
> > **[Part 2/3] Thank you for your thorough review and valuable feedback! We address all questions raised by the reviewer.**
> >
> > **Q5:** *Q5 How can we rule out the possibility that CIC is just a better way to pre-train a DDPG agent in these settings w.r.t. other baselines? This would be significant anyway, but do the authors believe that the same results would generalize to different base algorithms (say TRPO, SAC, A2C...)?*
> >
> > The choice of using a DDPG to optimize CIC was simply to be consistent (in terms of optimization backbone) to the baselines we compare to used in URLB. We do not expect major differences between DDPG / SAC or other off-policy algorithms, since there is nothing DDPG-specific in CIC design decisions. However, there may be a difference between intrinsic reward computation (for any exploration algorithm, not just CIC) between off-policy (DQN, DDPG, SAC) and on-policy algorithms (TRPO, PPO).
> >
> > Since off-policy algorithms sample from all data collected by the agent throughout its lifetime, they are closer to the IID assumption than on-policy methods which only use information from most recent trajectories which are highly correlated. For this reason, we would expect off-policy explorations algorithms to be generally more stable since the samples used for intrinsic reward computation are less correlated. This observation is not specific to CIC but a general observation about computing intrinsic rewards for unsupervised RL algorithms in off-policy vs on-policy settings.
> >
> > **Q6:** *Do the authors believe that the combination of DDPG and CIC would be successful in different settings (e.g., visual discrete domains such as Atari games) as well?*
> >
> > We believe CIC could be successfully applied to other domains because it is independent of the RL optimization algorithm. We only used a DDPG to be consistent with the setup in URLB. CIC addresses fundamental limitations of prior methods as shown in our experiments (main results, ablations in Fig 6, and Appendices I & K) and does not rely on any domain specific inductive biases, so we believe it would also work in other domains with other RL optimization algorithms. We focused on URLB because it was explicitly designed for unsupervised RL and has baselines for all common unsupervised RL algorithms - knowledge, data, and competence-based - allowing us to evaluate CIC not only with respect to other competence-based methods but unsupervised RL algorithms as whole.
> >
> > **Q7:** *From my understanding, the empirical results are not directly comparable with the URLB (Laskin et al., 2021) despite a very similar setting.*
> >
> > Thank you for asking for this clarification. While URLB is relatively recent, our results are directly comparable to the ones in the official URLB benchmark, since we use the same code for optimization (was open-sourced and publicly available on OpenReview during the NeurIPS 2021 submission cycle https://anonymous.4open.science/r/urlb). Our evaluation is also the same (2M pretraining, 100k finetuning steps), and all the baselines and CIC are trained with the same DDPG optimization backbone provided by URLB. The only difference between CIC and the baselines in terms of code is the intrinsic reward specification. Specifically, we built CIC using the code open-sourced by URLB for other competence-based algorithms (DIAYN, SMM, APS).
> >
> > **Q8:** *The results section seems to imply that a key factor under the improvement over previous competence-based methods is the ability of CIC to cope with a larger skill space. Can the authors clarify why previous methods are prevented to work with a comparable skill space?*
> >
> > Thank you for your question. We motivate this in Section 3, which we’ve re-titled as “Motivation” to make this clearer, since we agree with you that the current title may have not been easily interpreted as a motivation section. Specifically we discuss the intuition for why prior methods have used small skill spaces in the paragraph titled “Why it is important to utilize high-dimensional skills”.
> >
> > The main reason that prior methods only accommodate small skill spaces is that the discriminator $q(z|\tau)$ is effectively a classifier that takes tau and classifies the skill $z$ (or vice versa depending on the decomposition). To accommodate a large skill dimension, this classifier needs an exponentially larger number of state samples. This is why prior works empirically used small skill spaces (DIAYN $16-100$ dim one-hot, APS $10$ dim continuous). With the contrastive discriminator CIC is able to accommodate much larger dimensions (e.g. $64$ continuous dim shown in Fig 6(b)).
> >
> > **Q9:** *The normalized score of Fig. 6 does not seem to match the one of Fig. 4;*
> >
> > Fig 4 shows results after finetuning the pre-trained policy for 100k steps. Fig 6 shows zero-shot performance (no fine-tuning of the policy). The reason is that Fig 6 consists of ablations of various parts of CIC (projection net, skill dim, etc) and zero-shot evaluation allows us to investigate the quality pre-training decoupled from finetuning.
> >
> > [continued]

---

> > > ### Author Response · Authors · 2021-11-14
> > > **[Part 3/3 + References] Thank you for your thorough review and valuable feedback! We address all questions raised by the reviewer.**
> > >
> > > **Q10:** *Can they also provide a comparison with previous methods when working with the same (potentially lower-dimensional) skill space?*
> > >
> > > This is a great question. We show this in Fig 6(b) by ablating the skill dimension for CIC, and find that a lower skill dimension for CIC results in much worse zero-shot performance on the downstream tasks. There’s roughly a $10\times$ difference between performance using low-dimensional skills ($5-10$ dims) and the optimal skill size ($64$ dims). This suggests that with lower dimensional skills ($5-10$ dims) CIC would likely perform on par with prior competence-based methods but not substantially better.
> > >
> > > **Q11:** *The novelty seems limited, as CIC is essentially similar to APS (Liu and Abbeel, 2021) with a different discriminator loss (which has been employed for unsupervised skills discovery before)*
> > >
> > > While we agree that the overall MI objective is similar to APS, we disagree that the discriminator loss “has been employed for unsupervised skills discovery before.”
> > >
> > > CIC is the first paper to propose using contrastive loss for the discriminator between $\tau$ and abstract latent skills $z$. Prior works have effectively classifiers to estimate $q(z|\tau)$ or $q(\tau|z)$. The reason for using contrastive (to support larger skills spaces) was not obvious and it also was not obvious how to implement this contrastive objective in practice, which we examine in this paper. This new contrastive discriminator is the main contribution of this work, and we believe it is novel in the sense that this work is the first to propose it, motivate it, and define a specific implementation that achieves state-of-the-art empirical results.
> > >
> > > There is a prior work (DISCERN) that employed a similar contrastive loss, but it did so in the context of visual goal-conditioned RL as opposed to unsupervised skill discovery. We provide a detailed comparison between CIC and the most closely related prior works in Appendix D.
> > >
> > > **Q12:** *It is not completely clear from the paper what are the specific factors that lead to such a performance improvement over previous works.*
> > >
> > > Thank you for asking for this clarification. The specific factor that leads to significant performance improvements is the new contrastive discriminator introduced in this work. There are two baselines that are closely related to CIC - namely, APT and APS. APT uses only the entropy term while APS uses the entropy term as well as a (classifier) discriminator.
> > >
> > > Given the gains over APT and APS, the performance improvement is most likely due to the new contrastive discriminator, since all other components of the algorithm are fixed. We argue and show experimental evidence for why contrastive discrimination is important (it accommodates large skill spaces where prior methods could not) in Fig. 6, Appendix I, and Appendix K. We hope that the strong empirical main results as well as these additional ablations are sufficiently convincing and provide intuition as to why the CIC objective works well in practice.
> > >
> > > **Q13:** *It is not easy to track the different baselines in Fig. 4 (top). What is this plot representing?*
> > >
> > >
> > > We borrow this presentation of results from Rliable (Agarwal et al. 2021). This figure is an aggregate plot showing target normalized scores on the x-axis and the fraction of runs (for each algo) that achieve a greater than or equal score to the target on the y-axis. A perfect algorithm would achieve a score of 1.0 (expert score) everywhere, so it would be a horizontal line at 1.0 on the plot. This figure is useful because various statistics can be derived from it. The area under the curve is the mean. The area above the curve is the optimality gap. The horizontal line at $0.5$ on the y-axis is the median.
> > >
> > > ### References
> > >
> > > **Rliable** Rishabh Agarwal, Max Schwarzer, Pablo Samuel Castro, Aaron Courville, and Marc G. Bellemare. Deep reinforcement learning at the edge of the statistical precipice, 2021.
> > >
> > > **URLB** Michael Laskin, Denis Yarats, Hao Liu, Kimin Lee, Albert Zhan, Kevin Lu, Catherine Cang, Lerrel Pinto, and Pieter Abbeel. Urlb: Unsupervised reinforcement learning benchmark, NeurIPS Benchmark Track, 2021
> > >
> > > **DADS** Archit Sharma, Shixiang Gu, Sergey Levine, Vikash Kumar, and Karol Hausman. Dynamics-aware unsupervised discovery of skills. ICLR, 2020.
> > >
> > > **RVIC** Kate Baumli, David Warde-Farley, Steven Hansen, Volodymyr Mnih, Relative Variational Intrinsic Control, arxiv:2012.07827, 2020
> > >
> > > **DISCERN** David Warde-Farley, Tom Van de Wiele, Tejas Kulkarni, Catalin Ionescu, Steven Hansen, and Volodymyr Mnih. Unsupervised control through non-parametric discriminative rewards, 2018.
> > >
> > > **CPC** Aaron van den Oord, Yazhe Li, and Oriol Vinyals. Representation learning with contrastive predictive coding. arXiv:1807.03748, 2018.
> > >
> > > **SimCLR** Ting Chen, Simon Kornblith, Mohammad Norouzi, and Geoffrey E. Hinton. A simple framework forcontrastive learning of visual representations. ICML, 2020.

---

> > > > ### Comment · Reviewer_VTZ1 · 2021-11-19
> > > > **Thank you for your replies**
> > > >
> > > > I would like to thank the authors for their thorough replies to my questions and concerns. Most of my previous doubts have been now cleared, and my bottom line for this work would be something like:
> > > >
> > > > *A conditional entropy estimator inspired by contrastive predictive coding allows to employ the forward MI decomposition and so to adopt larger skill vectors, which leads to a great performance in the URLB.*
> > > >
> > > > I think this is a significant finding, even if a thorough theoretical understanding of why these MI approaches actually work is still to find, especially if the performance is not clearly correlated to tighter estimates of the MI. Anyway, I will consider raising my score upon discussions with other reviewers, while I suggest the authors to incorporate some their clarifications in the main paper (Q1 and Q10 in particular).
> > > >
> > > > I report below some additional questions that the authors might address (note that this are minor questions that will not meaningfully affect my evaluation of the paper).
> > > >
> > > > > This is a great question! The reason we (and prior works) must use a parametric estimation of MI is that we need to backpropagate through $z$ in order to learn a skill space that will produce useful exploratory behaviors. A fully nonparametric estimate of MI would not allow us to backpropagate through $z$.
> > > >
> > > > Can you clarify why a non-parametric MI estimate would prevent from backpropagating the loss to $z$?
> > > >
> > > > > We do not expect major differences between DDPG / SAC or other off-policy algorithms, since there is nothing DDPG-specific in CIC design decisions.
> > > >
> > > > Let me rephrase the question here: The architecture and the way to optimize it (DDPG) have been fixed for the evaluation, which might bias the result in some (hidden) way. The ordering of the algorithms' performance would be preserved under changes of the architecture and the optimization procedure?

---

> > > > > ### Author Response · Authors · 2021-11-23
> > > > > **Thank you for the follow up questions. We clarify them below.**
> > > > >
> > > > > Thank you for your response! We have updated the paper with clarifications (for Q1 see Appendix M; for Q10 see end of page 8). We clarify your follow-up questions below:
> > > > >
> > > > > **Q1:** *Can you clarify why a non-parametric MI estimate would prevent from backpropagating the loss to $z$?*
> > > > >
> > > > > A non-parametric MI estimate would introduce a non-differentiable operation. Since we want to backpropagate back through the z encoder, we want the MI (or if not the full MI then just the discriminator) to be differentiable. You could, of course, use approximate gradients through a non-differentiable operation using straight-through estimator or Gumbel-Softmax but this would add unnecessary complexity and a potential source of instability.
> > > > >
> > > > > **Q2:** *Let me rephrase the question here: The architecture and the way to optimize it (DDPG) have been fixed for the evaluation, which might bias the result in some (hidden) way. The ordering of the algorithms' performance would be preserved under changes of the architecture and the optimization procedure?*
> > > > >
> > > > > This is an interesting question. While it is of course possible that the fixed DDPG evaluation might bias CIC in some hidden way, it is unlikely. One reason is that the same argument can be applied to any of the baselines since CIC and baseline derivations / implementations are independent of the optimization algorithm. Another data point is that we use the same hyperparameters for the DDPG as those suggested by the URLB authors  (other than skill dimension $\text{dim}(z)$ and weighting coefficient $\alpha$), so the DDPG was not even tuned for CIC. It is, of course, possible that CIC somehow got lucky with optimal hyperparameters out-of-the-box, but this is a highly unlikely scenario in deep RL research. The ordering is therefore most likely to be the same regardless of which off-policy algorithm is used.
> > > > >
> > > > > The only place where CIC can benefit from off-policy learning more than other methods is in the large batch-size regime. It is possible that with larger batch sizes, CIC can leverage the additional negatives to get a more accurate estimate of the discriminator $q(\tau|z)$. However, this would be a favorable property of the algorithm. We keep the batch size fixed to the same value as used in URLB for the baselines ($B=1024$ for state-based unsupervised RL in URLB).
> > > > >
> > > > > Finally, we note that the DDPG optimization was used for a fair comparison with the baselines, and we used the exact DDPG code from URLB. So there was no CIC-specific decision made at any point when choosing the underlying optimization algorithm.
> > > > >
> > > > > We hope these answers address your remaining concerns. Thank you again for spending time to reviewer our work!

---

### Official Review · Reviewer_cP9K · 2021-10-30

**Correctness:** 3
**Technical Novelty And Significance:** 2
**Empirical Novelty And Significance:** 4
**Recommendation:** 8
**Confidence:** 4

**Main Review:**

Strengths:

The empirical evaluation carried out in the paper is extensive with many baseline algorithms for skill-discovery from each class (knowledge-based, data-based, competence-based) on the recently proposed URLB benchmark. Further, evaluation metrics (IQM, Optimality gap etc.) used to measure performance are adopted from the recommendations in [1].

The discussion and analysis on the reasons for failure of current competence-based skill discovery algorithms. Ablation experiments (in Figure 6) help justify the design choices made by the CIC algorithm.

Weaknesses:

The proposed algorithm seems like a variation on an existing algorithm [2]. It refines some of the practical design choices used in the general framework of competence-based algorithms for unsupervised skill discovery. However, the authors have shown the differences between the various algorithms (in Table 1) and discussed their pros & cons which is useful to contextualize their contributions.

A few questions I have for the authors on the high-level motivations behind their algorithm design and would like some clarification on:

1] If we intuitively think of the notion of a skill as a form of abstraction of long-term behavior. For example motion primitives like walking, flipping etc. as shown in the paper which occurs over maybe a few tens or hundreds of steps. So, why do several skill discovery algorithms use only highly localized information in the state space (various tau instantiations such as single states or in this case single state-transitions (s, s’)) to infer its corresponding skill latents? Isn’t it more intuitive to infer these latents from more “global” quantities like entire episodes of policy rollouts. Could the authors comment on this?

2] To maximize the mutual information between state transitions and skills as defined In the CIC, we need to maximize the first term ($H[\tau]$) and minimize the second term ($H[\tau | z]$). This conditional entropy would be minimal when the corresponding distribution $p(\tau | z)$ is sharp/narrow (ideally like a delta-like density function) over the state-transition space. This seems rather counterintuitive to me. If we think of a single skill latent, say walking, shouldn’t the density $p( \tau | Z=z_{walk})$ have a high value for all the possible state-transitions of the walking primitive, which would be a rather wide distribution. Wouldn’t the pressure to keep this distribution as narrow as possible over the state-transition space given a single skill latent lead to several latents codes which essentially cover the same underlying behavior (redundant copies of different walking styles). Wouldn’t this be undesirable for generalization on downstream tasks which would require composing these unsupervised skills?

3] The authors argue for the need for increasing the dimensionality of the skill latents to ensure skill can be decoded back to a diverse set of behaviors. Couldn’t this be achieved by largely retaining the small latent spaces used for skills in prior work and using a more expressive policy decoder. This could allow for greater representation flexibility when the skill latents are decoded back to the action-space and ensure that skill latents give rise to a diverse set of behaviors.

Writing/Presentation:

The paper on the whole is well-written and easy to follow.

I found the phrasing in this sentence rather confusing and ambiguous. “If the set of behaviors outnumbers the set of skills, this will result in degenerate skills -- when one skill maps to multiple different behaviors”. What does “behaviors” refer to in this context -- action trajectories? If so, isn’t it expected that the set of skills would be much smaller (essentially a compressed representation) than the total number of “unique” action sequences. Unless, the authors

A few minor typos I found are below:

“Why most competence-base algorithms …. ” -> “competence-based algorithms … ”
“Both CIC and CIC use ...” -> do you mean CIC and APS?
In Algorithm 1 -> “Contrastive Intrinisc Control” -> “Intrinsic”

[1] Agrawal et.al, “Deep Reinforcement Learning at the Edge of the Statistical Precipice”, NeurIPS 2021.

[2] “APS: Active Pretraining with Successor Features”, ICML 2021.


**Summary Of The Paper:**

The paper proposes an algorithm (Contrastive Intrinsic Control) for unsupervised skill discovery by maximizing mutual information between skill latents and state transitions. The proposed algorithm is a refinement over existing methods [2]. It uses a contrastive method to estimate conditional entropy and measures entropy on state-transitions as opposed to simply states as done in previous methods [2]. The proposed algorithm shows good performance gains compared to existing competence-based skill discovery algorithms. Further, the paper also contains a rather extensive empirical evaluation of various skill discovery algorithms on the recently proposed URLB benchmark.


**Summary Of The Review:**

Although the proposed algorithm CIC is a variation on an existing algorithm [2], it shows impressive performance gains over several existing algorithms on a large suite of continuous control tasks from the URLB benchmark. These large-scale empirical evaluations and analysis of several algorithms for unsupervised skill discovery methods on a standard benchmark such as URLB would benefit the community.

---

> ### Author Response · Authors · 2021-11-14
> **[Part 1] Thank you for reviewing our work and your thought provoking questions! We address all questions below.**
>
> We thank the reviewer for their time and valuable feedback. We really enjoyed the insightful questions - they really hit at the core of what’s important in competence-based exploration, so thank you for asking these! We've corrected the typos you pointed out in the revised manuscript and provide answers / clarifications to your questions below:
>
> **Q1:** *why do several skill discovery algorithms use only highly localized information in the state space ... to infer its corresponding skill latents?*
>
> The reason most skill discovery papers (including CIC) use local information is that they operate under the assumption of a fully observed MDP. For this reason local information is sufficient to describe global behaviors (e.g. you don’t need to keep around memory of the past). However, in POMDPs where the environment is partially observed or highly multimodal environments where prior context is informative of future behavior we would definitely want to embed longer trajectory sequences.
>
> This is a very promising direction for future work to build on top of CIC. In CIC we investigated the specific problem of why competence-based methods perform poorly even in fully observed MDPs (like state-based URLB). Now that we have a working competence-based method in this setting, it is natural to extend this to harder partially observed environments. However, as observed in URLB, there is currently a big gap between state-based and pixel-based performance for all exploration algorithms, so there may be a representation-learning-for-exploration issue that needs to be solved first.
>
> **Q2:** *Wouldn’t the pressure to keep this distribution as narrow as possible over the state-transition space given a single skill latent lead to several latents codes which essentially cover the same underlying behavior?**
>
> This is a very insightful question! We can answer it by considering two limits (i) very large skill dimension (ii) very small skill dimension. (i) If we have a very large skill dimension, then we can overfit the discriminator by finding a 1:1 mapping between a skill and a state transition, and hence simply memorize states. This would result in not very useful skills as you point out. (ii) If we have a very small skill dimension, then the discriminator will be forced to map multiple behaviors into one skill. Since discrimination is coupled with exploration, this will result in a policy that will likely forget the harder skills in order to maximize the accuracy of the discriminator (this is what happens with methods like DIAYN).
>
> Now consider the in-between limit, where the skill dimension is not too small and not too large. Recall that the CIC objective also has an entropy term $H(\tau)$ that wants to maximize behavioral diversity. Since the skill dimension is not too large, it can’t simply memorize each transition, nor can it force the policy to collapse to static behaviors due to the $H(\tau)$ term. It is therefore forced to learn a wider distribution so that similar transitions fall into the same skill, although alone $-H(\tau|z)$  wants to fit a tighter distribution.
>
> Empirically, this may explain the behavior of the skill dimension ablation in Fig 6(b). In the small skill dimension limit, the policy cannot acquire diverse skills. Then it’s performance grows with the skill dimension until reaching a critical dimension of 64. Finally, at dimension 128 performance degrades again, possibly because the discriminator is starting to memorize transitions.
>
> **Q3:** *The authors argue for the need for increasing the dimensionality of the skill latents to ensure skill can be decoded back to a diverse set of behaviors. Couldn’t this be achieved by largely retaining the small latent spaces used for skills in prior work and using a more expressive policy decoder.*
>
> This is also a very interesting question - thank you for asking it!
>
> Rather than allowing the policy to decode to a diverse set of actions, the primary purpose of a large skill dimension is to produce diverse behaviors in the first place through the intrinsic objective. The intrinsic reward in eq (6) has two terms $H(\tau)$ and $-H(\tau|z)$. The problem if the skill space is too small is that the discriminator $-H(\tau|z) \geq \log q(\tau|z)$ wants to be accurate, and if there are only a small number of skills, then it will force the agent to produce a limited set of behaviors that are easier to classify. Of course, the term H(tau) should, in principle, keep up the behavioral diversity of the policy but as we see in Fig. 5, the discriminator dominates the intrinsic reward later in training (though importantly $H(\tau)$ is still non-zero). So we primarily need a large skill space for $q(\tau|z)$ to be able to map $z$ to diverse $\tau$ rather than $\pi(a|s,z)$ decoding to diverse $a$. This observation is also supported by the gridworld thought experiment in Appendix I.
>
> [continued]

---

> > ### Author Response · Authors · 2021-11-14
> > **[Part 2] Thank you for reviewing our work and your thought provoking questions! We address all questions below.**
> >
> > **Q4:** *What does “behaviors” refer to in this context -- action trajectories? If so, isn’t it expected that the set of skills would be much smaller (essentially a compressed representation) than the total number of “unique” action sequences.*
> >
> > “Behaviors” refers to state trajectories (or just state transitions in our case since we make the MDP assumption), since we care about how well the agent explores the state space. Yes you’re right that the set of skills should be much smaller than the set of possible trajectories, but it can’t be too small. As shown in Appendix I, in a `10x10` gridworld if you only have 4 skills, your behaviors will only cover a small fraction of the total environment. However, if you have 100 skills then you could simply memorize each state, and then your skills aren’t very useful either. So you don’t want your skill space to be too big either. However, we argue that current methods (DIAYN, SMM, APS, and so on) all suffer from the “too small” problem, and show how CIC can resolve this.
> >
> > Indeed, in Fig. 6(b) we show how CIC performance improves as you increase the skill dimension (up to 64 dims) but then degrades once you reach 128 dim skills. An interpretation for this phenomenon is that at 128 dim we start approaching the “too big” skill-space problem where our skills are just memorizing individual transitions and for that reason skill-conditioning is no longer as helpful. So you’re right, the skill space needs to be smaller than the observation space but not too small.

---

> > > ### Comment · Reviewer_cP9K · 2021-11-29
> > > **Response to Authors' Rebuttal**
> > >
> > > I'd like to thank the authors for their detailed response to my questions. Their responses helped clarify most of my concerns.

---

### Official Review · Reviewer_h3mC · 2021-11-02

**Correctness:** 3
**Technical Novelty And Significance:** 3
**Empirical Novelty And Significance:** 3
**Recommendation:** 8
**Confidence:** 3

**Main Review:**

The empirical evaluation is rigorous in comparison to common practice (120 runs and IQM stabilization) with strong SOA and baselines.

Liked the structured review of prior art that organizes work in an interesting thematic way.

I liked the empirical decomposition of reward into entropy and discriminator terms. It is interesting that latent state entropy is important early in learning and task discrimination is more important later on.

Useful to know that higher dimensional skills are important (64D performed best).

Technically I(T;Z)  =  H(T) - H(T|Z)  =  H(Z) - H(Z|T) so theoretically it should not matter which way it is decomposed. The way the terms are calculated, however, could have a significant effect on the practical performance.

It isn’t clear what a “simple grid sweep of skills over the interval [0,1]” means.

Early stopping does not “leak information” so much as stop wasted exploration in irrelevant parts of the state space?

Typo: “we use particule” -> particle

The text refers to optimality gap, but the figures see to use expert normalized score which I assume is algorithm performance / DDPG baseline performance.

Did not fully understand the implications of the noise and projection argument, but it seems like an important and worth while design decision to investigate.

**Summary Of The Paper:**

The paper builds upon the DIAYN idea (Eysenback 2018) that an agent could develop skills in an unsupervised environment by finding a set of skills that collectively visits the whole state space but encourages each skill to cover a different subspace and later use one of these skills to simplify the learning of a downstream task. In this paper skills are learned in an unsupervised way using a mutual information based objective. The mutual information between latent states T and skill vector Z, I(T;Z), is decomposed as I(T;Z)=H(T)-H(T|Z) which the paper argues leads to explicit maximization of diversity of latent states T as well as distinct skills with focused effects by penalizing with H(T|Z). More diversity equals better exploration of more distant states leading to more interesting behaviors. The paper explains this decomposition also allows them to user higher-dimensional skill vectors that improve representational capacity and downstream performance. The paper develops a method to calculate the terms of this decomposition and formally shows it is a lower bound for the true mutual information. The latent state entropy H(T) is estimated using an unnormalized k-nearest neighbor method requiring an ad hoc scaling factor \alpha and the conditional entropy H(T|Z) is calculated using an NCE supervised neural net.  The paper illustrates the method by using their loss function to pretrain a DDPG architecture on unsupervised scenarios and then showing a benefit on downstream tasks. The proposed method, CIC, shows a larger IQM stabilized expert normalized score compared to state of the art methods.  Experiments also show that high-dimensio

**Summary Of The Review:**

The paper proposes a new algorithm for unsupervised behavior learning that is rigorously shown to be more effective and clearly argues for its design choices through supplementary experiments.

---

> ### Author Response · Authors · 2021-11-14
> **Thank you very much for reviewing our work! We address all questions below.**
>
> Thank you for spending time on reviewing CIC and providing us with valuable feedback! We’re glad that you liked the structure of the paper and found the results compelling. We corrected the typo you pointed out and provide clarifications to the reviewer’s questions below:
>
> **Q1:** *It isn’t clear what a “simple grid sweep of skills over the interval [0,1]” means.*
>
> Thank you for pointing this out, we’ve updated the “Skill architecture and adaptation ablations” subsection in Sec. 7 to make this clearer. A grid sweep is a simple skill selection strategy where you select skill values by sweeping over the interval $[0,1]$ and set all values in the skill vector to that value. For example, $z_i=0,  0.1, … , 0.9, 1.0$ would sweep over 11 different skills setting $z_i$ to the selected value for all $i$ in $z$. There are of course better strategies given sufficient data, but since we use a budget of only 4K steps to select skills and then 96K to finetune the weights (in order to be in line with the URLB evaluation of 100k finetuning steps), this simple strategy performs better relative to more sophisticated strategies such as CEM. We show this as an ablation in Fig 6(c).
>
> **Q2:** *Early stopping does not “leak information” so much as stop wasted exploration in irrelevant parts of the state space?*
>
> We agree that one interpretation is to stop wasted exploration in irrelevant parts of the state space, but we see this as “leaking information” in the sense that early termination tells the agent which parts of the state space are relevant to explore for the downstream tasks.
>
> For example, in Appendix K Figure 11 we run a simple ablation on the hopper from OpenAI mujoco gym vs the one from DeepMind control (DMC). OpenAI gym resets when the hopper loses balance, whereas DMC episodes have fixed length and do not reset early. We find that on OpenAI gym, a simple "Fixed" baseline where the agent receives a reward of $1.0$ for each timestep (and no other intrinsic or extrinsic rewards) substantially outperforms DIAYN and ICM in terms of zero-shot performance on the extrinsic reward of the task. This is because the hopper extrinsic reward has a term for standing up and one for moving. By maximizing the length of the episodes the simple “Fixed” baseline learns to stand reliably. This is therefore inadvertently leaking info that standing is a good thing to do for the downstream task.
>
> **Q3:** *The text refers to optimality gap, but the figures use expert normalized score which I assume is algorithm performance / DDPG baseline performance.*
>
> We use eval statistics defined in Rliable (Agarwal et al. 2021). In that work, the optimality gap is a statistic that measures how far the agent is from expert performance. Therefore the mean, median, and IQM statistics use expert performance (higher is better) while the optimality gap is a new statistic introduced in Rliable that measures how far away the agent is from expert performance (lower is better).
>
> We use the same definition of expert performance as in URLB (Laskin, Yarats et al. 2021) which is the score achieved by a supervised DDPG agent after 2M steps of training with access to extrinsic rewards. The expert baseline gets 2M reward interactions whereas baseline unsupervised RL algorithms and CIC only get 100k interactions with the extrinsic reward during finetuning.
>
> **Q4:** *Did not fully understand the implications of the noise and projection argument, but it seems like an important and worth while design decision to investigate.*
>
> Skill projection refers to the projection network (red box in Fig. 3). To compute the contrastive loss, we can either (i) pass in the raw skill sampled from noise as the key or (ii) project the skill with a neural network onto the key. This is a small difference in implementation, but leads to significant difference in performance. This is likely because it’s easier for the contrastive loss to discriminate in the projected space than the raw skill space (which is purely random). This observation is similar to the empirical gains observed from using the projection head in SimCLR and self-supervised siamese networks in general (e.g. SimSiam).
>
> ### References
>
> **Rliable** Rishabh Agarwal, Max Schwarzer, Pablo Samuel Castro, Aaron Courville, and Marc G. Bellemare. Deep reinforcement learning at the edge of the statistical precipice, 2021.
>
> **URLB** Michael Laskin, Denis Yarats, Hao Liu, Kimin Lee, Albert Zhan, Kevin Lu, Catherine Cang, Lerrel Pinto, and Pieter Abbeel. Urlb: Unsupervised reinforcement learning benchmark, NeurIPS Benchmark Track, 2021
>
> **SimCLR** Ting Chen, Simon Kornblith, Mohammad Norouzi, and Geoffrey E. Hinton. A simple framework for contrastive learning of visual representations. In International conference on machine learning, 2020.
>
> **SimSiam** Xinlei Chen, Kaiming He, Exploring Simple Siamese Representation Learning, arxiv:2011.10566, 2020

---

### Official Review · Reviewer_tKs2 · 2021-11-02

**Correctness:** 2
**Technical Novelty And Significance:** 2
**Empirical Novelty And Significance:** 3
**Recommendation:** 3
**Confidence:** 3

**Main Review:**

**Strengths**

* The experiments are done on URLB, which provides a good evaluation scheme for unsupervised RL. Well-established and common evaluation schemes are important for assessing methods empirically.
* The empirical performance, which is provided with relevant statistics, is good compared to multiple baseline methods.
* The analysis on the effect of different choices of multiple hyperparameters (Fig.6).

**Weaknesses**

* This work's originality is somewhat limited. Particle-based entropy maximization with state representations trained using a contrastive learning scheme has been explored in APT (Liu & Abbeel, 2021a). Using the same entropy maximization form for skill discovery was done by APS (Liu & Abbeel, 2021b).
* The motivation for using noise-contrastive estimation is not entirely clear. While the authors make a comparison with CPC as a lower bound of mutual information, CPC is not very commonly used in skill discovery and the usual variational lower bound can already be tight if the variational approximation $q$ approximates the true distribution $p$ perfectly.
* Theorem 1 is not technically correct because of the particle-based entropy term in $F_{CIC}$, which is also supported by the authors in that they introduced the weighting hyperparameter $\alpha$ due to that it doesn't consider the proportionality constant (Sec.4.2).
* The authors employ $I(\tau; z) = \mathcal{H}(\tau) - \mathcal{H}(\tau|z)$ as the decomposition of the mutual information (the 2nd line in Sec.4.1), but they use $q(z|\tau)$ instead of $q(\tau|z)$ for the rest of Sec.4.1.
* I think the derivation of the noise-contrastive estimator in Sec.4.1 needs more details. For instance, if there are noise samples, where do the correct samples come from?
* Using the notation $\tau$ for something other than trajectories can be misleading.
* Lack of empirical analysis with different values of $\alpha$.

**Summary Of The Paper:**

To tackle the unsupervised skill discovery problem, the authors attempt to maximize the mutual information between (latent) skills and states $I(\tau; z)$ by using the nonparametric particle-based entropy estimation for $\mathcal{H}(\tau)$ and noise-contrastive estimation for $\mathcal{H}(\tau|z)$. They evaluate their method and the baselines on the Unsupervised Reinforcement Learning Benchmark (URLB), where the agents are pre-trained without extrinsic rewards and fine-tuned for downstream tasks with extrinsic rewards and fewer environment steps.

**Summary Of The Review:**

While the empirical results basically show that CIC can outperform multiple baselines on URLB with an appropriately tuned value of $\alpha$, I am mainly concerned about the correctness of the claims and the novelty of the work.

---

> ### Author Response · Authors · 2021-11-14
> **[Part 1] Thank you for reviewing our work! We address all questions and concerns raised by the reviewer.**
>
> Thank you for your time and constructive feedback reviewing CIC! We’re glad that you found the evaluation to be well motivated and the empirical performance to be strong relative to the baselines. We address each of the posed questions below and revised the manuscript to incorporate the suggested clarifications:
>
> **Q1:** *Particle-based entropy maximization with state representations trained using a contrastive learning scheme has been explored in APT (Liu & Abbeel, 2021a).*
>
> While particle-based entropy maximization has been used before (Liu & Abbeel, 2021a), the contrastive learning scheme in CIC is new and substantially different from the one used in APT. In APT, the authors use the visual contrastive learning scheme introduced in CURL (Laskin, Srinivas et al. 2020) to learn embeddings that are invariant to data-augmentations (random crop) of the input image. In our work, we use contrastive learning to learn behaviors by maximizing agreement between latent skill vectors and state-transition embeddings.
>
> We’ve added pseudocode to the manuscript (Appendix L) for a detailed comparison between the two objectives - the main takeaway is that the CIC contrastive loss operates over trajectories $\tau$ and latent skill vectors $z$ (which has not been done before) while APT contrastive loss operates over data augmentations $\text{aug}(o)$ of the input image $o$ (as in CURL). These are two different contrastive objectives.
>
> **Q2:** *Using the same entropy maximization form for skill discovery was done by APS (Liu & Abbeel, 2021b).*
>
> While the high-level mutual information objective is the same between CIC and APS, the main contribution of CIC is to use a better discriminator (with contrastive behavior learning) as discussed in Q1. APS uses an MSE regression over successor features for the discriminator for $q(\tau|s)$ while CIC uses a new contrastive discriminator. Given the large performance difference between the two methods, the gains from CIC are due to the new contrastive objective introduced in this work, as this is the only difference between CIC and APS.
>
> **Q3:** *The motivation for using noise-contrastive estimation is not entirely clear.*
>
> We motivate CIC in Section 3 of the manuscript which was titled *“Distilling Diverse Behaviors into Distinct Skills.”* We apologize that we didn’t name the section more clearly, and have incorporated your feedback by changing the section title to *“Motivation.”* Specifically, the motivation for contrastive learning is explained in the subsection titled *“Why it is important to utilize high-dimensional skills”*. In this subsection, we argue that skill-based exploration needs to support large skill spaces but prior works use discriminators for $q(z|\tau)$ or $q(\tau|z)$ that can only support small skill spaces.
>
> Additionally, since noise contrastive learning works well for visual representation learning (e.g. SimCLR), it's interesting to leverage its power to learn skill conditioned representations. Different from prior work (APT and CURL) which use contrastive learning to learn representations from two augmented images, our representations are learned by skill based contrast. We hope this provides sufficient motivation for CIC.
>
> In the paper, these motivations are presented in Sections 3 and 4 and supported by results in Fig 6(b), Appendix I, and Appendix K.
>
> **Q4:** *Theorem 1 is not technically correct because of the particle-based entropy term in F_CIC ... due to that it doesn't consider the proportionality constant*
>
> We apologize for our notation, which we now realize was confusing. Theorem 1 is technically correct, but you’re right that our notation could have been clearer. The entropy particle estimator $H_\text{particle} (\tau)$ introduced in (Singh et al. 2003) is an asymptotically unbiased and consistent estimator for the entropy and includes the proportionality constant, so equation (3) is technically correct.
>
> However, we agree with you that we should use better notation in equation (6) which describes the intrinsic reward used for the practical algorithm. In equation (6) we use $H_\text{particle} (\tau)$ to refer to the unnormalized entropy estimator. We have edited equation 6 with updated notation to avoid this confusion. For a practical algorithm, we follow APT and ProtoRL and use a quantity proportional to $H_\text{particle} (\tau)$ rather than the exact entropy, which is simpler to compute in practice.
>
> **Q5:** *The authors employ  as the decomposition of the mutual information (the 2nd line in Sec.4.1), but they use  instead of  for the rest of Sec.4.1.*
>
> Thank you for pointing this out. This is a typo, the decomposition should always be using $q(\tau|z)$ and indeed this is what is used in the objective and the code we submitted. We have corrected this typo throughout the manuscript.
>
> [continued in Part 2]

---

> > ### Author Response · Authors · 2021-11-14
> > **[Part 2 + References] Thank you for reviewing our work! We address all questions and concerns raised by the reviewer.**
> >
> > **Q6:** *I think the derivation of the noise-contrastive estimator in Sec.4.1 needs more details. For instance, if there are noise samples, where do the correct samples come from?*
> >
> > Thank you for your question. We use the standard trick in contrastive learning where we sample a minibatch of size `B` of state transitions and skills, compute the inner products between the embeddings as described in Fig 3 and second term of eq (6). This gives us a `(B, B)` matrix of inner products. The terms along the diagonal are positives while the terms on the off-diagonal are negatives. We then pass this into a categorical cross entropy loss. We have added pseudocode to the appendix (see Appendix L) to make it clear how positives / negatives are sampled. This has been the standard way to compute contrastive losses from randomly sampled minibatches - see SimCLR (in vision) and CURL (in RL).
> >
> > **Q7:** *Using the notation tau  for something other than trajectories can be misleading.*
> >
> > Anticipating that using tau may be a bit confusing to the general RL community, we explicitly define it in our Notations section. We agree that it is not ideal, but it is the notation that the unsupervised RL community has adopted. Specifically, we follow the notation used in the unsupervised RL ICML tutorial (Srinivas & Abbeel 2021). The reason is that all competence based algorithms use the same structure and estimate either $I(\tau;z) = H(z) - H(z|\tau)$ or  $I(\tau;z) = H(\tau) - H(\tau|z)$ and differ mostly in their different choices for $\tau$, which can be a single state, a transition, a starting state and goal state, or a trajectory. Since all of these are special cases of a trajectory, the community has adopted the notation $\tau$.
> >
> > **Q8:** *Lack of empirical analysis with different values of alpha*
> >
> > As described in Section 4 of the paper and clarified in Q4, the hyper-parameter $\alpha$ is just there to make the practical algorithm simple by avoiding the need to compute the proportionality term for the particle entropy. As such, we choose $\alpha$ to roughly weigh the entropy and contrastive terms equally. The raw values in Fig 6 show that choosing $\alpha=0.9$ results in approximately equal weighting and we found that this performed quite well in practice. The only exception was the Jaco arm environment, where $\alpha=0$ resulted in the best performance. We explain why $\alpha=0$ worked best for the Jaco arm in Sec. 7 - the tl;dr is that the $H(\tau)$ term is not necessary in that environment, only the discriminator.
> >
> > ### References
> >
> > **APT** Hao Liu and Pieter Abbeel. Behavior from the void: Unsupervised active pre-training. arXiv preprint arXiv:2103.04551, 2021a.
> >
> > **CURL** Michael Laskin, Aravind Srinivas, and Pieter Abbeel. Curl: Contrastive unsupervised representations for reinforcement learning. In International Conference on Machine Learning, 2020b.
> >
> > **APS** Hao Liu and Pieter Abbeel. APS: active pretraining with successor features. In International Conference on Machine Learning, 2021b.
> >
> > **Entropy particle estimation** Harshinder Singh, Neeraj Misra, Vladimir Hnizdo, Adam Fedorowicz, and Eugene Demchuk. Nearest neighbor estimates of entropy. American Journal of Mathematical and Management Sciences, 23(3-4):301–321, 2003.
> >
> > **ProtoRL** Denis Yarats, Rob Fergus, Alessandro Lazaric, and Lerrel Pinto. Reinforcement learning with prototypical representations. In International Conference on Machine Learning, 2021b.
> >
> > **ICML Unsupervised RL Tutorial** Aravind Srinivas and Pieter Abbeel. Unsupervised learning for reinforcement learning, 2021. URL https://icml.cc/media/icml-2021/Slides/10843_QHaHBNU.pdf.
> >
> > **SimCLR** Ting Chen, Simon Kornblith, Mohammad Norouzi, and Geoffrey E. Hinton. A simple framework for
> > contrastive learning of visual representations. In International conference on machine learning, 2020.

---

> > > ### Comment · Reviewer_tKs2 · 2021-11-21
> > > **Thank you for the very detailed response. Here are my follow-up comments.**
> > >
> > > I appreciate the authors for providing the long and thorough comment and updating the manuscript in response to the initial reviews.
> > >
> > > I would like to ask the authors for additional elaboration on the following.
> > >
> > > **Q4**
> > >
> > > Actually, the proportionality constant wasn't the only issue I was concerned about.
> > >
> > > > The entropy particle estimator $H_\text{particle} (\tau)$ introduced in (Singh et al. 2003) is an asymptotically unbiased and consistent estimator
> > >
> > > Yes, the particle-based entropy estimator is asymptotically unbiased. If I'm not mistaken, it means that we cannot say that the (of course, imperfect) estimator $\mathcal{H}_\text{particle} (\tau)$ itself is always equal to or smaller than the true $\mathcal{H} (\tau)$ (unlike the frequently used, variational lower bound's case). The Proof of Theorem 1 just uses $\mathcal{H} (\tau)$ instead of $\mathcal{H}_\text{particle} (\tau)$. If we were to assume perfect estimators, can't we replace many tight lower bound approximations with equalities in existing literatures?
> > >
> > > **Q5 and Q6**
> > >
> > > > This is a typo, the decomposition should always be using $q(\tau|z)$ and indeed this is what is used in the objective and the code we submitted.
> > >
> > > I hoped it was a typo, but it still doesn't seem clear if it was (please see my following comments).
> > >
> > > Could you clarify $q(\tau_i|z_i) = \exp\left[\frac{1}{N}\sum_{i=1}^N \left( f(\tau_i,z_i) - \log \frac{1}{N} \sum_{j=1}^N \exp(f(\tau_i, z_j)) \right) \right]$ with answers to the following points?
> > >
> > > * What exactly is the number of samples $N$? Is it the number of all possible elements in the sample space of $\tau$ and $z$, or is it the same as the minibatch size $B$? While the authors are right that subsampled negatives (i.e., negative samples from minibatches) are used in contrastive objectives *for training*, I believe it should be the former here to satisfy the equation above. But then, the same $N$ is used in Eq.(6), the practical definition of the intrinsic reward, which I find unclear.
> > > * More importantly, according to the equations (above and from the manuscript) and the attached source code, the authors employ different $z_j$'s as negative samples and compute the cross entropy loss. However, doesn't it result in estimating $q(z|\tau)$ instead of $q(\tau|z)$? In other words, in order to learn $q(\tau|z)$, shouldn't it use different $\tau_j$'s as negative samples with fixed $z$?

---

> > > > ### Author Response · Authors · 2021-11-23
> > > > **[Part 1/2] Thank you for the follow up questions! We clarify all the points raised below.**
> > > >
> > > > Thank you for the follow up questions! We finally understand the source of confusion (see Q3 below - entirely our fault, now fixed). We clarify below.
> > > >
> > > > **Q1:** *If we were to assume perfect estimators, can't we replace many tight lower bound approximations with equalities in existing literatures?*
> > > >
> > > > The reviewer is right that the proof of Theorem 1 is independent of the specific estimator used for $H(\tau)$. Theorem 1’s sole purpose is to motivate our objective (Entropy + CPC) as opposed to just using CPC. Theorem 1 is estimator agnostic for entropy - it just says that with $H(\tau)$ we get a tighter bound than just CPC.
> > > >
> > > > Of course, for a practical algorithm we need to choose an estimator. In principle - yes you can use any estimator for $H(\tau)$ or MI in literature. We choose the variational estimator for MI and particle estimator for $H(\tau)$ because it makes it easiest to compare CIC to prior work. Current SOTA data-based exploration are methods that use a particle estimator to maximize entropy (APT, ProtoRL). Prior competence-based methods (DIAYN, SMM, APS) all use variational estimators for MI. Additionally, the best performing competence-based method (APS) also uses the particle estimator for $H\(tau)$ which allows us to isolate where CIC gains are coming from (i.e. the discriminator). CIC is therefore directly comparable to all these prior works, since the only change is the new contrastive discriminator $\log q (\tau |z)$.
> > > >
> > > > You’re completely right that other forms of entropy estimation would be interesting to investigate, but that is not the focus of this work. The focus of CIC is showing that competence-based approaches to exploration are significantly improved with a contrastive discriminator and the $I(\tau;z) = H(\tau) - H(\tau|z)$ decomposition is chosen similar to APS to encourage exploration through $H(\tau)$. Our aim was to leverage powerful contrastive methods that have recently emerged in vision to learn better skill embeddings, and our empirical results confirm that this is a useful method for unsupervised skill discovery that substantially improves on prior work. Specifically, since the main difference between APS and CIC is the discriminator, this suggests that the gains are coming from the contrastive estimation of $-H(\tau|z)$.
> > > >
> > > > **Q2:** *What exactly is the number of samples $z$? Is it the number of all possible elements ... or is it the same as the minibatch size $N$?*
> > > >
> > > > Thank you for following up on this. We now see the source of confusion - we used $N$ to denote both negatives and batch size (which is the same in our case practically, but the two can be different in theory). For this reason, we’ve revised eq. (3)-(6) to make this clearer.
> > > >
> > > > Practically, the number of negatives is the same as the minibatch size. As you note “the authors are right that subsampled negatives (i.e., negative samples from minibatches) are used in contrastive objectives for *training*.” We only need intrinsic reward for training and for this reason can sample negatives from the minibatch.
> > > >
> > > > Perhaps the misunderstanding is that the reviewer thinks the intrinsic reward is computed during environment interaction (when no negatives are available)? If so, this is not the case. The intrinsic reward is computed during the gradient update step. This makes sense because the same transition $(s,s’)$ can have different values for $r_{\text{int}}$ over the course of training, and is how $r_{\text{int}}$ is computed for all algorithms in URLB, not just CIC.
> > > >
> > > > The intrinsic reward is only needed when we update the Q function by minimizing the Bellman loss. During training we sample a minibatch B of $(s,a,s’,d)$ transitions (notice no rewards) to compute the actor / critic losses. We then compute the $r_{\text{int}}$ using these transitions. We therefore always have negatives (other samples in the minibatch) when computing $r_{\text{int}}$.
> > > >
> > > > Then during downstream adaptation, the agent fine-tunes using the extrinsic reward, which does not require negatives. Our main results are measurements of extrinsic rewards during evaluation after 100k steps of fine-tuning.

---

> > > > > ### Author Response · Authors · 2021-11-23
> > > > > **[Part 2/2] Thank you for the follow up questions! We clarify all the points raised below.**
> > > > >
> > > > > **Q3:** *More importantly, according to the equations (above and from the manuscript) and the attached source code, the authors employ different $z$'s as negative samples and compute the cross entropy loss.*
> > > > >
> > > > > Thank you for pointing this out. In the code used to generate the results for this work, negatives were sampled with fixed $z$, since we were doing the $I(\tau;z) = H(\tau) - H(\tau|z)$ decomposition. When we cleaned up the code for submission, it looks like we introduced this typo by accident. We fixed this typo (see newly uploaded code). Then, to sense-check we re-ran 3 seeds per evaluation task (12 tasks with 3 seeds each) and found that it matches the reported results - see results here https://sites.google.com/view/iclrcic/home.
> > > > >
> > > > >
> > > > > When cleaning / refactoring code for submission we accidentally swapped the queries / keys, which is equivalent to fixing $\tau$ and sampling $z$ negatives instead of fixing $z$ and sampling $\tau$ negatives. We've fixed this and have uploaded the correct version of the code (one line change swapping queries / keys back so that query <-> skills, key <-> transitions). To sense-check it we did the following (you can see tables / plots on this website https://sites.google.com/view/iclrcic/home) :
> > > > >
> > > > > 1. We re-ran the main experiments reported in the paper (pre-training for 2M steps with CIC, then finetuning for 100k steps) with the revised codebase with 3 seeds per evaluation task. Our results match those reported in the paper. The **paper reported a mean 0.78 normalized score** across 10 seeds per task. The **codebase achieved a mean normalized score of 0.82** with 3 seeds per task, confirming that our results are consistent with those reported in the paper.
> > > > > 2. We also re-ran pre-training with the old codebase used to generate results for this paper for the walker domain and compared read-out statistics to pre-training with the submitted codebase (with typo fixed). We find that the pre-training statistics (e.g. intrinsic rewards, cpc contribution, entropy contribution) are very similar.
> > > > >
> > > > >
> > > > > Independently, there's a notational mistake which started in eq (3) and was then copy / pasted into subsequent equations and above responses. You're correct that the sum over negatives in eq (3) and subsequent equations should be over $\tau$ with fixed $z$. Otherwise, Theorem 1 does not hold. For instance, eq. (5) which just states the lower bound from CPC needs to sum over $\tau$ negatives to be consistent with eq (4). We very much appreciate you noticing these notational errors. We've fixed them but can understand how the initial submission was confusing! Thank you for pointing this out!

---

### Decision · Program_Chairs · 2022-01-20

**Decision:**

Reject

**Comment:**

The paper addresses the question of skill discovery in reinforcement learning: can we (without supervision) discover behaviors so that later (when supervision is available via a reward signal) we can learn faster? The paper proposes a new contrastive loss that an agent can optimize for this purpose, based on a decomposition of mutual information between skills and transitions. The reviewers praised the extensive experimental evaluation and good empirical results, as well as the analysis of failure modes of related algorithms.

Unfortunately, there appeared to be errors in the derivation and implementation. (These include typos in derivations that made them difficult to follow, as well as uploaded code that didn't match the experimental results.) While the authors claim to have fixed all of them, the reviewers were not all completely convinced by the end of the discussion period. In any case, these errors caused confusion during review; so, whether the errors are fixed or not, it seems clear that there hasn't been time for a full evaluation of the corrected derivations and code. For this reason, it seems wise to ask that this paper be reviewed again from scratch before being published.